# Safe Opponent-Exploitation Subgame Refinement

**Mingyang Liu**[*1], **Chengjie Wu**[*1], **Qihan Liu**[2], **Yansen Jing**[1], **Jun Yang**[2],
**Pingzhong Tang**[1], **Chongjie Zhang**[1]
[1]Institute for Interdisciplinary Information Sciences, Tsinghua University
[2]Department of Automation, Tsinghua University
`{liumy19, wucj19, lqh20, jingys19}@mails.tsinghua.edu.cn`
`{yangjun603, kenshin, chongjie}@tsinghua.edu.cn`

## Abstract

In zero-sum games, an NE strategy tends to be overly conservative confronted with opponents of limited rationality, because it does not actively exploit their weaknesses. From another perspective, best responding to an estimated opponent model is vulnerable to estimation errors and lacks safety guarantees. Inspired by the recent success of real-time search algorithms in developing superhuman AI, we investigate the dilemma of safety and opponent exploitation and present a novel real-time search framework, called Safe Exploitation Search (SES), which continuously interpolates between the two extremes of online strategy refinement. We provide SES with a theoretically upper-bounded exploitability and a lower-bounded evaluation performance. Additionally, SES enables computationally efficient online adaptation to a possibly updating opponent model, while previous safe exploitation methods have to recompute for the whole game. Empirical results show that SES significantly outperforms NE baselines and previous algorithms while keeping exploitability low at the same time.

## 1 Introduction

Behind the recent breakthroughs of superhuman AIs in Go [Silver et al., 2016, 2017, Schrittwieser et al., 2020], heads-up no-limit Texas hold'em (HUNL) [Brown et al., 2018, Moravčík et al., 2017, Brown and Sandholm, 2019, Brown et al., 2020], and Hanabi [Lerer et al., 2020], search plays a vital role. In perfect information games, Monte Carlo tree search (MCTS) is widely applied to improve policy's strength. In zero-sum imperfect information games such as poker, search algorithms are used to find a Nash equilibrium (NE) approximation in subgames encountered in real time [Brown and Sandholm, 2017, Burch et al., 2014]. They are both theoretically sounded and empirically powerful.

In zero-sum games, NE-based search algorithms [Burch et al., 2014, Moravcik et al., 2016, Brown and Sandholm, 2017, Brown et al., 2018] find safe strategies with low exploitability and produce strong baselines against all opponents [Brown and Sandholm, 2019]. However, it may be overly conservative confronted with opponents with limited rationality, and fail to take advantage of their weaknesses to obtain higher rewards [McCracken and Bowling, 2004, Johanson et al., 2007, Li and Miikkulainen, 2018]. From another perspective, there have been extensive studies on opponent exploitation to address the problem. Some typical works [Carmel and Markovitch, 1996, Billings et al., 2003, Gilpin and Sandholm, 2006, Li and Miikkulainen, 2018] model the opponent's strategy based on previous observations and then search for a new strategy to exploit this model. However, these methods often neglect the significance of the strategy safety, thus being highly exploitable.

Few exceptions including Johanson et al. [2007] and Ganzfried and Sandholm [2015a] aim to search for safe and robust counter-strategies. Ganzfried and Sandholm [2015a] provides a characterization of

---

[*]Equal contribution

36th Conference on Neural Information Processing Systems (NeurIPS 2022).

safe deviations from NE in repeated games. Restricted Nash response (RNR) [Johanson et al., 2007] finds a Pareto optimal strategy with respect to safety and exploitation in the full game. However, it is computationally inefficient because it needs to recompute a strategy for the whole game whenever the opponent model is updated. This can be even infeasible in an online setting where the opponent model is being updated continuously with streamed data.

In this paper, we study the dilemma of **safety** and **opponent exploitation** and present a new scalable real-time search framework **Safe Exploitation Search (SES)** that continuously interpolates between the two extremes of strategy search, hence unifying safe search and opponent exploitation. It enables computationally efficient online adaptations to a continuously changing opponent model, which is hard to address by previous safe exploitation algorithms. The safety criterion requires the refined strategy to stay close to NE, formally speaking, to expose limited exploitability against any opponents, while the opponent exploitation criterion requires the strategy to adapt to its specific opponent and to exploit its weaknesses. We propose a novel maximization objective in the subgame search framework which combines the safety objective and exploitation, controlled by the exploitation level $\alpha$. We construct a new gadget game to optimize this objective, which enables our method's scalability to large games such as Texas Hold'em. Theoretically, we prove that SES is guaranteed to outperform NE at the cost of some constant increase in its own exploitability confronted with non-NE opponents.

Empirically, we evaluate the effectiveness of our search algorithm in 1 didactic matrix game 2 poker games: *Leduc Hold'em* [Southey et al., 2005] and *Flop Hold'em Poker* (FHP) [Brown et al., 2019]. The experiment results demonstrate that our algorithm significantly outperforms NE baselines against non-NE opponents and keeps low exploitability at the same time. Additionally, we show that SES is not only much more computationally efficient than previous safe exploitation methods but also more robust to estimation errors in opponent models.

## 2 Related work

This paper investigates the problem of safe opponent exploitation in two-player zero-sum imperfect information games. We propose a novel search algorithm that balances between NE and exploiting opponents. Two major relevant research areas are search algorithms in imperfect information games and opponent exploitation.

**Search in imperfect information games.** In recent literature, search techniques are witnessed to be important in developing strong AI strategies in both perfect and imperfect information games [Burch et al., 2014, Moravcik et al., 2016, Brown and Sandholm, 2017]. Texas hold 'em poker is widely employed as a benchmark for imperfect information games. A primary part of the long-term research on Texas hold'em poker is the evolution of subgame solving algorithms, which aim at achieving a more accurate Nash equilibrium approximation in the subgame encountered given a pre-computed strategy for the full game which we refer to as the blueprint strategy. Unsafe search [Billings et al., 2003, Ganzfried and Sandholm, 2015b, Gilpin and Sandholm, 2006, 2007] estimates the subgame reach probability assuming the opponent follows blueprint, and searches for a refined subgame strategy. Subgame resolving [Burch et al., 2014] and maxmargin search [Moravcik et al., 2016] are theoretically sounded **safe** search algorithms which ensure that the subgame strategy obtained is no worse than the blueprint. They search in a gadget game and achieve safety by providing the opponent with the option not entering the current subgame. DeepStack [Moravčík et al., 2017] and Libratus [Brown et al., 2018] build strong poker AIs with the aid of search. Beyond poker, search algorithms for subgame refinement have also shown promise in improving joint strategies in cooperative imperfect information games such as Hanabi [Lerer et al., 2020] and the bidding phase of contract bridge [Tian et al., 2020]. The purpose of our search algorithm is different from previous methods in poker literature. We seek to exploit opponents while keeping exploitability low, rather than simply approximating NE.

**Opponent exploitation.** Most previous opponent exploitation researches [Carmel and Markovitch, 1996, Billings et al., 2003, Gilpin and Sandholm, 2006, Li and Miikkulainen, 2018] typically model the opponent's strategy based on previous observations and then search for a new strategy to exploit this model, but put little emphasis on safety.

One similar work is Johanson et al. [2007] which proposes $p$-restricted Nash response (RNR) to find a safe exploitation strategy to the estimated opponent's strategy. It calculates a Nash equilibrium for the whole game restricting that the opponent plays the estimated strategy $\sigma^{\text{fix}}$ with probability $p$,

and any strategy with probability $1 - p$. In that paper, Johanson et al. [2007] prove that a $p$-RNR to $\sigma^{\text{fix}}$ is Pareto optimal with respect to exploitation and safety. However, it does not provide an explicit bound. Additionally, whenever the estimated opponent model changes or we want to use a different $p$ to balance between safety and exploitation, the original $p$-RNR has to recompute the strategy for the whole game. It is computationally inefficient in an online setting, where the opponent model is updated after every round with new game data. Our algorithm instead takes modeling error into account and provides explicit bounds for both safety and exploitation. With the aid of real-time search, it only searches for strategies in subgames encountered instead of the whole game.

Ganzfried and Sandholm [2015a] study safe exploitation strategies in repeated games, which is a different setting from this paper. Intuitively, it achieves safety by risking in exploitability at most what it has earned over NE in expectation in previous rounds. Therefore, its expected value in the whole repeated game is never worse than the NE. In contrast, this paper focuses on the safety of stage game strategies. Furthermore, our algorithm is complementary to Ganzfried and Sandholm [2015a]. Ganzfried and Sandholm [2015a] calculate an $\varepsilon$-safe best response for the whole game at each iteration with LP. This procedure is one of the main limitations on the algorithm's scalability. Our algorithm can be a possible efficient substitute for the calculation.

Bernasconi-de Luca et al. [2021] use a UCB-like algorithm to learn how to encourage opponent's engagement in repeated games setting by guaranteeing that the opponent's utility lies in the desired range. Moravcík et al. [2017] use a similar mixing distribution technique as ours to speed-up resolving procedure for NE by fixing the distribution of infosets on top of the subgame with a prediction of some unknown NE strategy. This work does not study opponent exploitation. Besides, it does not provide a safety or an exploitation bound for the algorithm.

To our knowledge, we are the first paper to investigate the safe opponent exploitation problem in subgame resolving schemes. Subgame resolving enables online adaptations to a continuously changing opponent model, eliminating the need to recompute a whole game strategy. It offers computational benefits in practical opponent exploitation circumstances. Our experiments in section 5 demonstrate its efficiency and robustness.

There is extensive research [Albrecht and Stone, 2018] on agent modeling. However, this paper only focuses on the safe exploitation algorithm, but not the agent modeling techniques. We can use off-the-shelf agent modeling algorithms to estimate the opponent's strategies.

## 3 Notations and background

An extensive-form imperfect information game $G = (P, H, Z, A, \chi, \rho, \cdot, \sigma_c, u, \mathcal{I})$ describes sequential interactions among agents, where agents have private information. A finite set $P$ consists of $n$ players and a chance node $c$ which represents the stochastic nature of the environment. The set of non-terminal decision nodes is denoted as $H$, and $Z$ is a set of terminal nodes or leaves. The set of possible actions is $A$, and $\chi : H \to 2^{|A|}$ is a function which assigns to each decision node $h \in H$ a set of legal actions. A player function $\rho : H \to P$ assigns to each decision node a player $p \in P$ who acts at that node. If action $a$ leads from $h$ to $h'$, we write $h \cdot a = h'$. If there exists a sequence of actions leading from $h$ to $h'$, we write $h \sqsubseteq h'$. At each node $h \in H$, the acting player $p = \rho(h)$ picks an action from legal actions $a \in \chi(h)$, and leads node $h$ into its child $h \cdot a$. The chance node always samples an action from its own distribution $\sigma_c$, which is common knowledge to all players. Utility functions are $u = (u_1, u_2, \ldots, u_n)$, where $u_i : Z \to \mathbb{R}$ defines the utility of player $i$ at terminal node $z \in Z$. The nature of imperfect information is characterized by infosets $\mathcal{I} = (\mathcal{I}_1, \mathcal{I}_2, \ldots, \mathcal{I}_n)$, where $\mathcal{I}_i = (I_{i,1}, \ldots, I_{i,k_i})$ is a partition of $H$ for player $i$. Two states in the same infoset must have the same acting player and the same legal action sets. We use $I(h)$ to denote the infoset that $h$ belongs to. A player $p$ cannot distinguish between states $h_1$ and $h_2$ if $I(h_1) = I(h_2)$, and thus should behave identically on all states in the same infoset.

The strategy of a player $p$ is $\sigma_p : \mathcal{I}_p \times A \to \mathbb{R}$, where $\sigma_p(I, a)$ is a distribution over valid actions on infoset $I$. For simplicity, we also use $\sigma_p(h, a)$ to denote $\sigma_p(I(h), a)$. We use $\pi^\sigma(h)$ to denote the probability of reaching state $h$ from the root when agents choose a strategy profile $\sigma = \langle \sigma_1, \sigma_2, \ldots, \sigma_n \rangle$. Formally, $\pi^\sigma(h) = \prod_{h' \cdot a \sqsubseteq h} \sigma_{\rho(h')}(h', a)$. We use $\pi^\sigma_{-p}(h) = \prod_{h' \cdot a \sqsubseteq h \wedge \rho(h') \neq p} \sigma_{\rho(h')}(h', a)$ to denote the probability of reaching $h$ when player $p$ always chooses the action that leads to $h$ whenever possible. $\pi^\sigma(h, h')$ is the reaching probability of $h'$ from $h$. $\pi^\sigma(h \cdot a, h')$ is the the probability of

reaching $h'$ from $h$ if action $a$ is taken at $h$. These probabilities can be formally defined in a similar manner. A **blueprint** is a pre-computed strategy for the full game.

The expected utility of player $p$ given strategy profile $\sigma$ is $u_p^\sigma = \sum_{z \in Z} \pi^\sigma(z) u_p(z)$. The **counterfactual value** $v_p^\sigma(I, a)$ is the expected utility that player $p$ will obtain after taking action $a$ at infoset $I$, given the joint policy profile is $\sigma$. Mathematically, it is the weighted sum of expected values at all states $h \in I$.

$$v_p^\sigma(I, a) = \frac{\sum_{h \in I, z \in Z} \pi_{-p}^\sigma(h) \pi^\sigma(h \cdot a, z) u_p(z)}{\sum_{h \in I} \pi_{-p}^\sigma(h)} \tag{1}$$

We further define $v_p^\sigma(I) = \sum_{a \in A} \pi_p^\sigma(h, a) v_p^\sigma(I, a)$.

In the rest of the paper, we focus on two-player zero-sum games with perfect recall. Zero-sum means $\forall z \in Z, u_1(z) + u_2(z) = 0$. Perfect recall means that no player will forget the information which has been obtained previously in the game. This is a common assumption in related literature.

A best response strategy $BR_p(\sigma_{-p}) = \arg\max_{\sigma_p} u_p^{\langle \sigma_p, \sigma_{-p} \rangle}$ for player $p$ is the strategy that maximize his own expected utility against fixed opponent strategy $\sigma_{-p}$. The **exploitability** of strategy $\sigma_p$ is $\exp(\sigma_p) = u_p^{\sigma^*} - u_p^{\langle \sigma_p, BR_{-p}(\sigma_p) \rangle}$ where $\sigma^*$ is the optimal strategy, and is an NE in two-player zero-sum games. It measures the performance of $\sigma_p$ against its best response comparing with the NE. A **counterfactual best response** $CBR_p(\sigma_{-p})$ is a strategy where $\sigma_p(I, a) > 0$ if and only if $v_p^\sigma(I, a) \geq \max_b v_p^\sigma(I, b)$. Counterfactual best response is a best response, but not vice versa. The **counterfactual best response value** $CBV_p^{\sigma_{-p}}(I) = v_p^{\langle CBR_p(\sigma_{-p}), \sigma_{-p} \rangle}(I)$ is the expected utility of the counterfactual best response policy. Since we focus on two-player zero-sum games, we will use $CBV^{\sigma_p}(I)$ as a shorthand notation for $CBV_{-p}^{\sigma_p}(I)$.

We follow the imperfect information subgame definition as in Burch et al. [2014]. An **augmented infoset** contains states which cannot be distinguished by the remaining players.

**Definition 3.1.** An imperfect information subgame $S$ is a forest of trees, closed under both the descendant relation and membership within augmented infosets for any player. Let $S_{\text{top}}$ be the set of nodes which are roots of each tree in $S$.

# 4 Method

In this section, we introduce our novel search algorithm called safe exploitation search (SES), which exploits the weaknesses of the opponent while ensuring a bounded exploitability efficiently. Let $\sigma$ be the pre-computed blueprint strategy. Without loss of generality, assume we search for player 2's refined strategy $\sigma_2^S$ by applying SES to all subgames $S \in \mathbb{S}$. Finally, the refined strategy for P2 after search is $\sigma_2'$, which is the same as $\sigma_2$ in $\{I_2^i | \forall S \in \mathbb{S}, I_2^i \notin S\}$ and is replaced with $\sigma_2^S$ in $S \in \mathbb{S}$.

## 4.1 Safe Exploitation Search

Our algorithm offers a unified approach to balance these two demands with theoretical guarantees. The objective of our search algorithm is to find a new subgame strategy $\sigma_2^S$ for $S \in \mathbb{S}$ which maximizes

$$\begin{aligned} SE(\sigma_2^S) = \alpha \sum_{I_1^j \in S_{\text{top}}} \hat{p}(I_1^j) \left( v_1^\sigma(I_1^j) - CBV_1^{\sigma_2^S}(I_1^j) \right) \\ + (1 - \alpha) \min_{I_1^j \in S_{\text{top}}} \left( v_1^\sigma(I_1^j) - CBV_1^{\sigma_2^S}(I_1^j) \right), \end{aligned} \tag{2}$$

where $\alpha \in [0, 1]$ is a hyper-parameter controlling the exploitation level, and $\hat{p}(I_1^j)$ is the estimated probability of player 1 entering infoset $I_1^j \in S_{\text{top}}$. Given P2's strategy (which is the blueprint $\sigma_2$) and P1's actual strategy (which does not have to be the blueprint $\sigma_1$), the real probability of player 1 entering infoset $I_1^j \in S_{\text{top}}$ (which we denote as $p(I_1^j)$) is determined. $\hat{p}(I_1^j)$ is an estimation of $p(I_1^j)$. For instance, in poker, it is the estimated distribution of private cards player 1 holds. Both theoretically and empirically, such estimation does not have to be fully accurate. It can be done with off-the-shelf opponent modeling techniques, which lies beyond the focus of this paper.

Intuitively, the maximization objective achieves a balance between **opponent exploitation** and **safety**, controlled by **exploitation level** $\alpha$. The first part of the objective is maximized when $\sum_{I_1^j \in S_{\text{top}}} \hat{p}(I_1^j) CBV_1^{\sigma_2^S}(I_1^j)$ is minimized. It aims at finding a strategy $\sigma_2^S$ which results in the lowest value for P1 under the assumption that the reach probabilities is $\hat{p}$. It can be interpreted as exploiting the estimated P1's strategy. The second part of the objective demands the resolved strategy to behave well against any reach probability distribution. We use the subgame margin $\min_{I_1^j \in S_{\text{top}}} \left( v_1^\sigma(I_1^j) - CBV_1^{\sigma_2^S}(I_1^j) \right)$ [Moravcik et al., 2016] which can be regarded as the worst-case utility increase for P2.

Our search objective is a convex combination of exploitation and safety, which is closely related to previous safe exploitation research [McCracken and Bowling, 2004, Johanson et al., 2007]. RNR [Johanson et al., 2007] calculates an exploitation strategy by computing an NE in the full game, restricting the opponent to play its fixed strategy with probability $p$ and any other strategy with $1 - p$. RNR is proved to be Pareto optimal with respect to safety and exploitation. However, it neither provides an explicit bound on exploitability and performance nor takes modeling errors into account. Furthermore, without search, RNR has to recompute for the whole game whenever the opponent's strategy changes, which limits its efficiency. Experiment section 5.3 demonstrates that SES is much more computationally efficient, and section 5.4 shows that, even we augment RNR with search framework, SES is still much more robust to estimation errors in opponent strategy.

By maximizing the objective 2, we provide sound theoretical results for both safety and opponent exploitation. Additionally, we provide analyses of how (1) exploitation level $\alpha$, (2) accuracy of opponent modeling, and (3) strength of the blueprint strategy impact the theoretical bound. By gradually increasing $\alpha$ from 0 to 1, our algorithm tends to exploit rather than keep safe.

**Theorem 4.1.** *(safety) Let $\mathbb{S}$ be a disjoint set of subgames $S$. Let $\sigma^* = \langle \sigma_1^*, \sigma_2^* \rangle$ be the NE where P2's strategy is constrained to be the same with $\sigma_2$ outside $\mathbb{S}$. Define $\Delta = \max_{S \in \mathbb{S}, I_1^i \in S_{top}} |CBV_1^{\sigma_2^*}(I_1^i) - v_1^\sigma(I_1^i)|$. Let $\tilde{p}(I_1^i)$ be the reach probability given by $\sigma_1^*$. Let $\hat{p}(I_1^i)$ be the estimation of reach probability $p(I_1^i)$ given by the real opponent strategy. Define $\tau = \max_{S \in \mathbb{S}, I_1^i \in S_{top}} |\frac{\hat{p}(I_1^i) - \tilde{p}(I_1^i)}{\tilde{p}(I_1^i)}|$. Whenever $1 - (2\tau + 1)\alpha > 0$, we have a bounded exploitability given by:*

$$\exp(\sigma_2') \leq \exp(\sigma_2^*) + \frac{2}{1 - (2\tau + 1)\alpha} \Delta. \tag{3}$$

Recall that $\sigma_2'$ is the refined strategy after search. The proof is provided in Appendix B. This theorem implies that the exploitability of the new strategy is smaller than that of strategy $\sigma_2^*$ plus a constant value, which is the closest strategy to NE if constrained to differ from $\sigma_2$ only in $\mathbb{S}$. The corresponding theoretical result of maxmargin search [Moravcik et al., 2016], a safe search algorithm with no opponent exploitation abilities, is $\exp(\sigma_2') \leq \exp(\sigma_2^*) + 2\Delta$. Comparing these two results, we can interpret the term $2/(1 - (2\tau + 1)\alpha)$ as the additional risk introduced by exploiting the opponent. If exploitation level $\alpha = 0$, then our bound is as tight as that of maxmargin search [Moravcik et al., 2016]. The bound also gets tighter if the $\tau$ gets smaller, or the blueprint $\sigma_2$ is closer to $\sigma_2^*$.

**Theorem 4.2.** *(opponent exploitation) Let $\epsilon = \|\hat{p} - p\|_1$ be the L1 distance of the distribution $p(I_1^i)$ and $\hat{p}(I_1^i)$. Let $\eta = \min_{S \in \mathbb{S}} \max_{I_1^j \in S_{\text{top}}} \left( CBV_1(I_1^j, \sigma_2^S) - CBV_1(I_1^j, \sigma_2^*) \right) \geq 0$. We use $BR_p^{[\mathbb{S}, \sigma_p]}(\sigma)$ to denote the strategy for player $p$ which maximizes its utility in subgame $S \in \mathbb{S}$ against $\sigma_{-p}$ under the constraint that $BR_p^{[\mathbb{S}, \sigma_p]}(\sigma)$ and $\sigma_p$ differs only inside $\mathbb{S}$. By maximizing objective 2, for all $S \in \mathbb{S}$, the refined strategy $\sigma_2'$ satisfies*

$$u_2^{\left\langle BR_1^{[\mathbb{S}, \sigma_1]}(\sigma_2'), \sigma_2' \right\rangle}(S) \geq u_2^{\left\langle BR_1^{[\mathbb{S}, \sigma_1]}(\sigma_2^*), \sigma_2^* \right\rangle}(S) + \frac{1 - \alpha}{\alpha}(\eta - 2\Delta) - \epsilon\eta \tag{4}$$

The proof is provided in Appendix B. Observe that the reach probability $p$ is characterized by P1's strategy outside $\mathbb{S}$ and $\hat{p}$ is its estimation. Because the search algorithm always find a stronger response strategy for P1 in $\mathbb{S}$ (which is exactly $BR_1^{[\mathbb{S}, \sigma_1]}(\sigma_2')$) as well, opponent exploitation refers to adapting to P1's strategy $\sigma_1$ outside $\mathbb{S}$. This theorem implies that the utility of the new strategy

$\sigma'_2$ is lower bounded by the utility of $\sigma^*_2$ when both confronted with P1's unknown strategy outside $\mathbb{S}$. It provides theoretical guarantees for the opponent exploitation ability of our algorithm. $\epsilon$ can be interpreted as estimation error. The lower bound increases if the estimation error get smaller or the blueprint $\sigma_2$ is closer to $\sigma^*_2$. We show empirically how exploitation level $\alpha$ and estimation error impact both safety and exploitation abilities in section 5.

## 4.2 Gadget Game

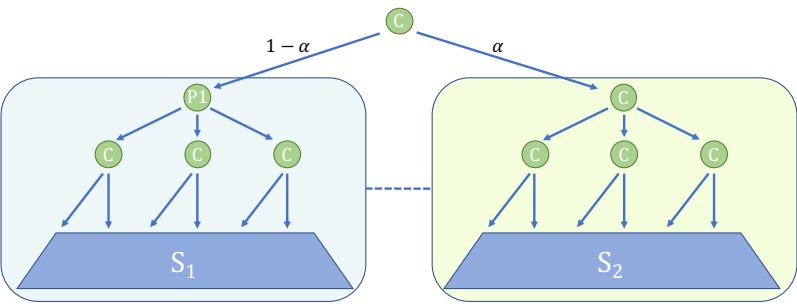

Figure 1: The gadget game of SES. The shadow and dashed line indicate that player 2 cannot distinguish between the two branches. $C$ represents chance node, $P1$ represents player 1's action node. $S_1$ and $S_2$ are two identical copies of the subgame $S$ with utility shifted.

In order to find $\sigma^S_2$ which maximize objective 2, a straight-forward method is to reformulate the maximization problem as a Linear Programming problem [Moravcik et al., 2016]. However, LP solvers [Koller et al., 1994] cannot handle large-scale problems. Alternatively, inspired by [Moravcik et al., 2016], we create a gadget game and then apply iteration-based NE algorithms such as CFR [Zinkevich et al., 2007, Tammelin et al., 2015, Lanctot et al., 2009] in the gadget game. The gadget game is carefully designed such that the NE solution found in it is exactly the solution to the original optimization problem.

As shown in Figure 1, the original subgame is copied into two identical parts $S_1, S_2$ in the gadget game. Player 2's infosets stretch over both branches, while player 1 can distinguish between the two parts. We use P1 to denote player 1, and P2 for player 2. The procedure of constructing such gadget game can be summarized into 4 steps as described below:

1. Create a chance node at the top of the gadget game. It goes to the left part with probability $1 - \alpha$, and the right part with probability $\alpha$. The outcome is visible to P1 but not P2. Therefore, corresponding nodes in both branches are in the same infosets for P2, and his strategy $\sigma^S_2$ will be the same for both parts. For P1, notice that S1 and S2 only differs in the distribution of the infosets of player 1. Given the strategies of player 2 in S1 and S2 are the same, the difference between the distribution of infosets of player 1 on the top of the subgame will not change the counterfactual value of player 1. Therefore, we will have the same counterfactual regret and the same strategies generated by CFR for player 1 in S1 and S2.

2. We subtract $u_1(z)$ by $v^\sigma_1(I^i_1)$ for all $z \sqsubseteq h, h \in I^i_1$ where $I^i_1$ denotes the infoset on the top of the subgame, and add $u_2(z)$ by $v^\sigma_1(I^i_1)$ in order to keep the subgame zero-sum on both S1 and S2. By doing so, the objective of $p_2$ will change from maximizing $-CBV^{\sigma^S_2}_1(I^i_1)$ to maximizing $v^\sigma_1(I^i_1) - CBV^{\sigma^S_2}_1(I^i_1)$.

3. As for the left part of the gadget game, the P1 node enables P1 to enter an arbitrary infoset $I^i_1$. The following chance nodes sample a specific state with probability proportional to $\pi^\sigma_{-1}(h)$ for all $h \in I_1$. Since this is a zero-sum game, in an NE strategy, he will enter the one with lowest $v^\sigma_1(I^i_1) - CBV_1(I^i_1, \sigma^*_2)$ which is exactly the minimization in the second term of $SE(\sigma^S_2)$.

4. The chance node in the right part will sample an infoset $I^i_1$ according to reach probability $\hat{p}(I^i_1)$. The following chance nodes again sample a specific state with probability proportional to $\pi^\sigma_{-1}(h)$ for all $h \in I_1$. So the NE objective of this part is exactly the summation in the first term of $SE(\sigma^S_2)$.

The pseudocode of SES is shown in Appendix A. We also provide a didactic matrix game in Appendix D as an example to show the necessity of considering safety and expected payoff simultaneously, and to demonstrate the superiority of SES over a simple mixing strategy.

# 5 Experiment

Our experiment is done in *Leduc Hold'em* [Southey et al., 2005] and *Flop Hold'em Poker* (FHP) [Brown et al., 2019]. *Leduc Hold'em* is a smaller-scale poker game and FHP is a larger one. The rules of these two pokers are provided in Appendix C. We demonstrate the exploitability and evaluation performance of SES against opponents of various strengths. The exploitability measures a search algorithm's safety, while head-to-head evaluation measures the ability of opponent exploitation. We also illustrate how estimation accuracy of opponent's strategy and the exploitation level $\alpha$ impact the results. Please refer to Appendix E for implementation details.

## 5.1 Opponents

In our experiments, we test the performance of our algorithm against opponents of various strengths. We create 3 types of opponents with 3 random seeds each. The first one is an approximation of NE, and is regarded as a strong opponent. For the second and third type of opponents, we enumerate every infoset in the blueprint strategy and shift the action distribution randomly with probability $\text{Pr}_{\text{shuffle}} = 0.3$ or $0.7$. We multiply the probability of each action by a random variable from $\text{Uniform}(0, 1)$, and then re-normalize the probability distribution. The procedure is motivated by Brown et al. [2018], in which such method is applied to create a number of diverse but reasonably strong agents. Even when $\text{Pr}_{\text{shuffle}} = 0.7$, the strategy keeps close to NE with average L1 distance of each infoset 0.132 comparing to 1.036 of a random strategy to NE. So they are regarded as opponents who are not fully rational but with competitive strength.

## 5.2 Safe Opponent Search

In Figure 2, we demonstrate the head-to-head evaluation performances and corresponding exploitability of the refined strategies found by SES against opponents of various strengths, under different exploitation level $\alpha$ and estimation errors of opponent's strategy. Different lines in each plot refers to corresponding estimation error $\epsilon$, which is the L1 distance of $\hat{p}$ and $p$. We evaluate our refined strategy when $\epsilon = 0.0, 0.3, 0.6, 0.9, 1.2$. Please refer to Appendix E for details of generating opponent estimations. The blue line is the result of blueprint strategy without conducting any search.

Generally speaking, SES balances between safety and opponent exploitation. The increase of exploitation level $\alpha$ helps win more chips from opponents, while resulting in the increase of the strategy's own exploitability. As can be seen in Figure 2, the exploitability increases when the exploitation level $\alpha$ grows from 0 to 1, which is consistent with Theorem 4.1. One exception is in FHP when $\epsilon$ is small: the exploitability surprisingly keeps decreasing even if SES puts more emphasis on opponent exploitation. Similar situations have also occurred in previous literature [Brown and Sandholm, 2017]. The reason is that our opponent is quite close to NE outside the subgame which will make $\hat{p}$ close to $\tilde{p}$ when $\epsilon$ is small, which means the $\tau$ in Theorem 4.1 is small. As a result, we will have a low-exploitability resolved strategy when using unsafe search and the exploitability increases as $\epsilon$ increases.

When the estimation is completely correct ($\epsilon = 0.0$, the yellow line), the expected payoff in FHP (3rd row in Figure 2) increases as the exploitation level $\alpha$ grows higher. In Leduc poker, since the game is very small, the pre-computed blueprint is very close to NE. Therefore, when confronted with relatively strong opponents ($\text{Pr}_{\text{shuffle}} = 0.0, 0.3$) which are also close to NE, actually few things can be done other than sticking with the blueprint. So the improvement introduced by SES is small. When facing relatively weak opponent ($\text{Pr}_{\text{shuffle}} = 0.7$), the improvement margin is slightly larger.

SES relies on an estimation of opponent's strategy. In order to test the robustness of our algorithm when the prediction of $p(I_1^i)$ is not accurate, we evaluate the performance of our algorithm with different values of estimation error $\epsilon$. As illustrated in Figure 2, the exploitability increases and the expected payoff drops when $\epsilon$ grows larger. The result is expected since an accurate estimation always provides benefits. However, it also demonstrates that SES can still achieve a trade-off between safety and opponent exploitation even when $\epsilon$ is considerably high. For instance, in FHP, $\epsilon$ is between 0 and

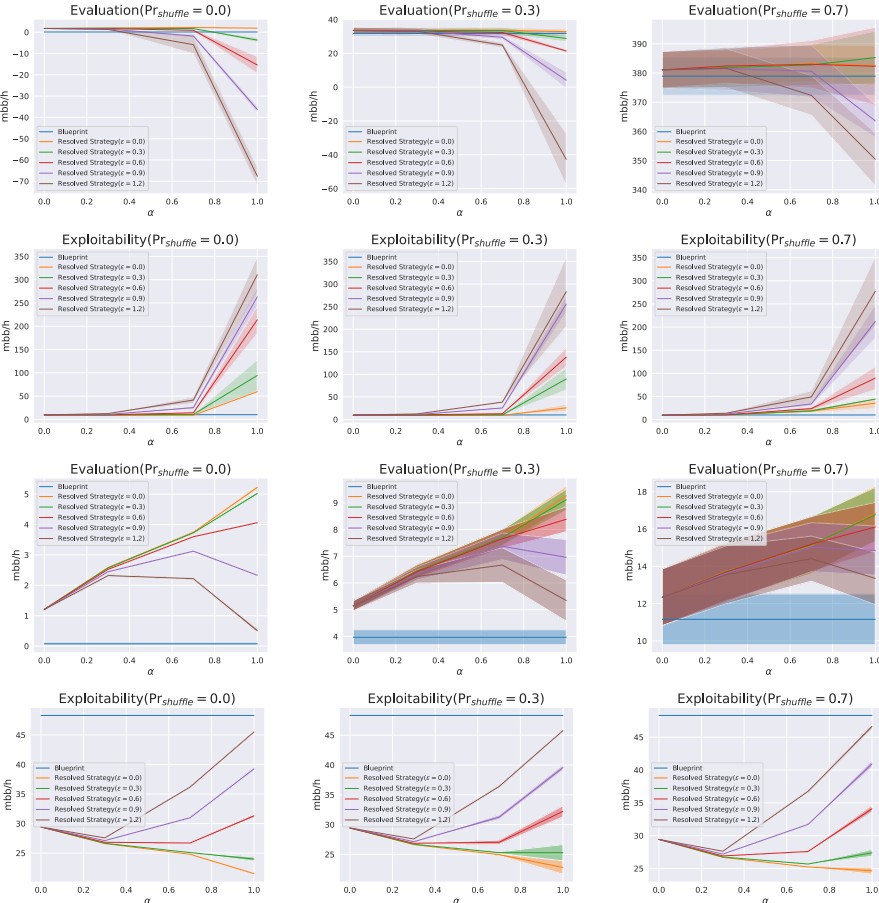

Figure 2: Experiment results on Leduc (Row 1 & 2) and FHP (Row 3 & 4). **Row 1 & 3:** Head-to-head payoffs against corresponding opponents. **Row 2 & 4:** Exploitability. Each row represents a type of opponent with $\text{Pr}_{\text{shuffle}} = 0.0, \ 0.3, \ 0.7$. The X-axis is the exploitation level $\alpha$.

2, and $\epsilon = 1.2$ means that the predicted distribution is almost random. When $\epsilon \leq 0.6$, the expected payoff still keeps increasing with respect to $\alpha$. In case of a bad estimation, we can always choose smaller $\alpha$ to ensure safety.

## 5.3 Comparison with Restricted Nash Response

We also compare SES with restricted Nash response (RNR) [Johanson et al., 2007], a previous safe exploitation algorithm, in FHP. RNR calculates an NE for for the whole game restricting that the opponent plays the estimated strategy $\sigma_{\text{fix}}$ with probability $p$, and any strategy with probability $1 - p$. In each round, we limit the computation time of RNR(normal) to 10 CPU second[*], which is the same for SES. However, as stated in section 4.1, RNR needs to recompute a strategy for the whole game in each round. It cannot converge in 10s. So we also compare with RNR(big), which has a budget of 10M CFR iterations in each round (around 190 CPU second in time). In contrast, SES only uses 10M CFR iterations to calculate its blueprint once. As is shown in Figure 3, SES significantly outperforms RNR(normal) in both exploitability and evaluation. SES also achieves much lower exploitability than RNR(big) and comparable evaluation results with much less computation time.

## 5.4 Comparison with EXP-STRATEGY

It is possible to augment $p-$RNR with real-time search. For a subgame $S$, we can create a similar gadget game to SES in Figure 1. The difference is that it keeps the opponent strategy fixed to its

---

[*]We test it on Intel(R) Xeon(R) Platinum 8276L CPU @ 2.20GHz

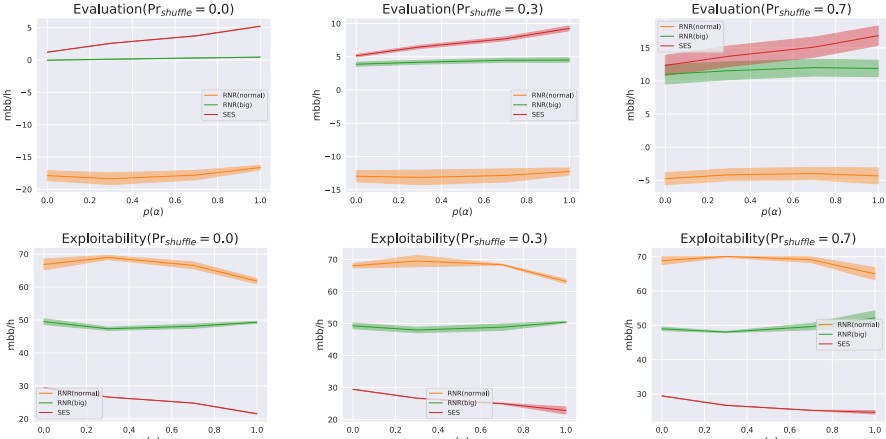

Figure 3: Comparison between SES and RNR. Each row represents a type of opponent with $\Pr_{\text{shuffle}} = 0.0, 0.3, 0.7$. The X-axis is the parameter $\alpha$ for SES and $p$ for RNR.

Table 1: Exploitability and evaluation performance of SES and EXP-STRATEGY. Strategy estimation error is $0.1$, and $\Pr_{shuffle} = 0.3$.

|  | $\alpha(p)$ | EXP-STRATEGY | SES |
|---|---|---|---|
| EXPLOITABILITY | 0.0 | 29.43($\pm$0.01) | 29.43($\pm$0.00) |
|  | 0.3 | 34.75($\pm$0.13) | 28.06($\pm$0.04) |
|  | 0.7 | 134.37($\pm$2.59) | 41.95($\pm$0.39) |
|  | 1.0 | 981.47($\pm$11.86) | 55.10($\pm$1.13) |
| EVALUATION | 0.0 | 5.15($\pm$0.19) | 5.15($\pm$0.19) |
|  | 0.3 | 1.63($\pm$0.40) | 6.21($\pm$0.25) |
|  | 0.7 | -27.22($\pm$0.91) | 6.00($\pm$0.81) |
|  | 1.0 | -88.94($\pm$6.07) | 3.30($\pm$0.96) |

estimation in the whole exploitation part, while SES only uses the reach probability $\hat{p}$ calculated from the estimated opponent strategy and allows the search algorithm to find opponent strategies in the exploitation part as well. To our best knowledge, this algorithm does not exist in previous literature, and can be regarded as an ablation study for SES. We call it EXP-STRATEGY.

Table 1 shows the exploitability and evaluation performance of SES and EXP-STRATEGY under different exploitation level $\alpha$ (or $p$ in RNR). We add errors in the estimation of opponent strategy, and use the same estimation for both algorithms. SES performs much better than EXP-STRATEGY: maintains lower exploitability and achieves higher evaluation performances[†] The experiment shows that EXP-STRATEGY is sensitive to modeling errors, because it relies on the estimated strategy in the whole subgame. SES, which exploits a distribution of infoset instead of the full opponent strategy, is more robust. Please refer to Appendix F for full comparison results.

Similar to SES, we derive the theoretical bounds for both exploitability and exploitation for EXP-STRATEGY. And the theory also demonstrates that EXP-STRATEGY is more sensitive to the accuracy of the estimation. Please refer to Appendix G for details.

## 6 Conclusion

We propose a novel safe exploitation search (SES) algorithm which unifies both safe search and opponent exploitation. With the aid of real-time search, SES can make online adaptations to a changing opponent model. We also prove safety and opponent exploitation guarantees of SES in Theorem 4.1 and Theorem 4.2. The experimental results in our designed matrix game confirm the

---

[†]The parameter $\alpha$ in SES and $p$ in RNR are not directly comparable. To be precise, we should compare the "frontier" of exploitability and exploitation for $\alpha, p \in [0, 1]$. For instance, under the same evaluation performance, which algorithm achieves lower exploitability. In table 1, SES is strictly better.

existence of the refined strategy which is both safe and actively exploiting the opponent. In games of poker, our method outperforms NE baselines while keeping exploitability low. SES is also much more efficient than previous safe exploitation algorithms without search. Additionally, SES is more robust to opponent modeling errors.

One limitation of SES is that it relies on an estimation of opponent. Although the assumption is common in opponent exploitation literature, efficient opponent modeling is still an active research area. Additionally, the exploitation level $\alpha$ is now regarded as a hyperparameter in SES. We find that $\alpha$ should be tuned for each specific purpose and automatically learning $\alpha$ is not trivial (see Appendix H for a brief discussion). We leave this for future work.

## Acknowledgments and Disclosure of Funding

The authors would like to thank the anonymous reviewers for their insightful discussions and helpful suggestions. This work is supported in part by Science and Technology Innovation 2030 – "New Generation Artificial Intelligence" Major Project (No. 2018AAA0100904) and National Natural Science Foundation of China (62176135).

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
