# A Algorithm

The pseudocode of SES in Poker is shown in Algorithm 1

---
**Algorithm 1** Procedure of SES in Poker
---
Initialize the game environment
Initialize the private cards distribution $p$ of player 2 as uniform distribution
Initialize the prediction of opponent private cards distribution
Compute a blueprint strategy $\langle \sigma_1, \sigma_2 \rangle$
**while** Game Not Terminate **do**
    $P \leftarrow$ current player
    **if** $P = P1$ **then**                                                          ▷ Opponent Player
        Observe opponent action $a$
        Update the prediction of opponent's private cards based on $a$
    **else if** $P = P2$ **then**                                     ▷ Resolving Player
        **if** Do Subgame Resolving on Current Node **then**
            Compute $v_1^\sigma(I_1^i)$
            Construct Gadget Game
            $\sigma_2^S \leftarrow CFR$(Gadget Game)
        **else**
            $\sigma_2^S \leftarrow \sigma_2$
        **end if**
        Sample action $a$ based on $\sigma_2^S$
        $p(I_2^i) \leftarrow p(I_2^i) \cdot \sigma_2^S(I_2^i, a)$
        Normalize $p$
    **else**                                                                  ▷ Chance Node
        Sample public cards $\mathcal{C} = \{c_1, c_2, ..., c_k\}$ from current chance node
        $\forall i = 1, 2, ..., k, p_{c_i} \leftarrow 0$
        Normalize $p$
    **end if**
**end while**
---

# B Proofs

## B.1 Proof of Theorem 4.1

**Theorem 4.1** (safety) Let $\mathbb{S}$ *be a disjoint set of subgames $S$. Let $\sigma^* = \langle \sigma_1^*, \sigma_2^* \rangle$ be the NE where P2's strategy is constrained to be the same with $\sigma_2$ outside $\mathbb{S}$. Define $\Delta = \max_{S \in \mathbb{S}, I_1^i \in S_{top}} |CBV_1^{\sigma_2^*}(I_1^i) - v_1^\sigma(I_1^i)|$. Let $\tilde{p}(I_1^i)$ be the reach probability given by $\sigma_1^*$. Let $\hat{p}(I_1^i)$ be the estimation of reach probability $p(I_1^i)$ given by the real opponent strategy. Define $\tau = \max_{S \in \mathbb{S}, I_1^i \in S_{top}} |\frac{\hat{p}(I_1^i) - \tilde{p}(I_1^i)}{\tilde{p}(I_1^i)}|$. Whenever $1 - (2\tau + 1)\alpha > 0$, we have a bounded exploitability given by:*

$$\exp(\sigma_2') \leq \exp(\sigma_2^*) + \frac{2}{1 - (2\tau + 1)\alpha}\Delta. \tag{5}$$

**Proof:**

For simplicity, we will omit the subscript of $CBV_1^{\sigma_2^*}$ by default. In order to prove Theorem 1, we will use mathematical induction on the level of the infoset. The depth $L$ has the same definition as in Brown and Sandholm [2017], i.e.

- For all the infosets which are direct parents of the subgames, we define $L(I) = 0$.

- For the infosets that are not ancestors of the subgames, we define $L(I) = 0$.

- For any infosets that are ancestors of the subgames, we define
  $L(I) = \max_{I' \in succ(I)} L(I') + 1$. That is, it has a higher level than any of its successors.

**Base Case of Induction**

Firstly, we will prove that for any infoset with level 0, the inequality of theorem 1 holds. For convenience, we consider that theorem 1 in a specific subgame $S$.

We will prove the infoset at the top of the subgame first. Since $SE(\sigma_2^*) \geq (1 - \alpha)(-\Delta) + \alpha \sum_i \hat{p}(I_1^i)(-\Delta) = -\Delta$, we have

$$(1 - \alpha) \min_{I_1^j} \left( v_1^\sigma(I_1^j) - CBV^{\sigma_2^S}(I_1^j) \right) + \alpha \sum_i \hat{p}(I_1^i)(v_1^\sigma(I_1^i) - CBV^{\sigma_2^S}(I_1^i))$$
$$= SE(\sigma_2^S) \tag{6}$$
$$\geq SE(\sigma_2^*)$$
$$\geq -\Delta$$

since $\sigma_2^S = \arg\max_{\tilde{\sigma}_2} SE(\tilde{\sigma}_2)$.

Furthermore, we have

$$\sum_i \hat{p}(I_1^i)(v_1^\sigma(I_1^i) - CBV^{\sigma_2^S}(I_1^i))$$
$$= \sum_i \hat{p}(I_1^i)(v_1^\sigma(I_1^i) - CBV^{\sigma_2^*}(I_1^i)) + \sum_i \hat{p}(I_1^i)(CBV^{\sigma_2^*}(I_1^i) - CBV^{\sigma_2^S}(I_1^i))$$
$$\leq \Delta + \sum_i \hat{p}(I_1^i)(CBV^{\sigma_2^*}(I_1^i) - CBV^{\sigma_2^S}(I_1^i)) \tag{7}$$
$$= \Delta + \sum_i \tilde{p}(I_1^i)(CBV^{\sigma_2^*}(I_1^i) - CBV^{\sigma_2^S}(I_1^i))$$
$$+ \sum_i (\hat{p}(I_1^i) - \tilde{p}(I_1^i))(CBV^{\sigma_2^*}(I_1^i) - CBV^{\sigma_2^S}(I_1^i))$$

where the second term $\sum_i \tilde{p}(I_1^i)(CBV^{\sigma_2^*}(I_1^i) - CBV^{\sigma_2^S}(I_1^i))$ is no larger than 0 because $\sum_i \tilde{p}(I_1)CBV^{\sigma_2^*}(I_1^i)$ is exactly what $\sigma_2^*$ minimized. Otherwise, $\sigma_2^*$ can change the strategy in the subgame so that he will get higher reward against $\sigma_1^*$ which conflicts the definition of NE.

And we will further decompose $I_1^i \in S_{top}$ into two parts, $\{I_1^{i,-}\}$ and $\{I_1^{i,+}\}$. They have the property that $CBV^{\sigma_2^*}(I_1^{i,-}) - CBV^{\sigma_2^S}(I_1^{i,-}) \leq 0$ and $CBV^{\sigma_2^*}(I_1^{i,+}) - CBV^{\sigma_2^S}(I_1^{i,+}) > 0$. And since $\sum_i \tilde{p}(I_1^i)(CBV^{\sigma_2^*}(I_1^i) - CBV^{\sigma_2^S}(I_1^i)) \leq 0$ as discussed above, we have

$$\sum_{I_1^{i,-}} \tilde{p}(I_1^i)(CBV^{\sigma_2^*}(I_1^{i,-}) - CBV^{\sigma_2^S}(I_1^{i,-})) + \sum_{I_1^{i,+}} \tilde{p}(I_1^i)(CBV^{\sigma_2^*}(I_1^{i,+}) - CBV^{\sigma_2^S}(I_1^{i,+}))$$
$$= \sum_i \tilde{p}(I_1^i)(CBV^{\sigma_2^*}(I_1^i) - CBV^{\sigma_2^S}(I_1^i)) \tag{8}$$
$$\leq 0$$

which implies that

$$\sum_{I_1^{i,+}} \tilde{p}(I_1^i)(CBV^{\sigma_2^*}(I_1^{i,+}) - CBV^{\sigma_2^S}(I_1^{i,+})) \leq -\sum_{I_1^{i,-}} \tilde{p}(I_1^i)(CBV^{\sigma_2^*}(I_1^{i,-}) - CBV^{\sigma_2^S}(I_1^{i,-})) \tag{9}$$

Then we have

$$\sum_i (\hat{p}(I_1^i) - \tilde{p}(I_1^i))(CBV^{\sigma_2^*}(I_1^i) - CBV^{\sigma_2^S}(I_1^i))$$

$$= \sum_{I_1^{i,-}} (\hat{p}(I_1^{i,-}) - \tilde{p}(I_1^{i,-}))(CBV^{\sigma_2^*}(I_1^{i,-}) - CBV^{\sigma_2^S}(I_1^{i,-}))$$

$$+ \sum_{I_1^{i,+}} (\hat{p}(I_1^{i,+}) - \tilde{p}(I_1^{i,+}))(CBV^{\sigma_2^*}(I_1^{i,+}) - CBV^{\sigma_2^S}(I_1^{I_1^{i,+}}))$$

$$\leq \tau \Big( -\sum_{I_1^{i,-}} \tilde{p}(I_1^{i,-})(CBV^{\sigma_2^*}(I_1^{i,-}) - CBV^{\sigma_2^S}(I_1^{I_1^{i,-}}))$$  (10)

$$+ \sum_{I_1^{i,+}} \tilde{p}(I_1^{i,+})(CBV^{\sigma_2^*}(I_1^{i,+}) - CBV^{\sigma_2^S}(I_1^{I_1^{i,+}})) \Big)$$

$$\leq -2\tau \sum_{I_1^{i,-}} \tilde{p}(I_1^{i,-})(CBV^{\sigma_2^*}(I_1^{i,-}) - CBV^{\sigma_2^S}(I_1^{I_1^{i,-}}))$$

$$\leq -2\tau \min_{I_1^j} \Big( CBV^{\sigma_2^*}(I_1^j) - CBV^{\sigma_2^S}(I_1^j) \Big)$$

The last inequation holds since $\min_{I_1^j} \Big( CBV^{\sigma_2^*}(I_1^j) - CBV^{\sigma_2^S}(I_1^j) \Big) \leq 0$ since $\sigma_2^*$ is the strategy with lowest exploitability by only changing strategy of $\sigma_2$ in the subgames.

Back to Equation 7, we have

$$\sum_i \hat{p}(I_1^i)(v_1^\sigma(I_1^i) - CBV^{\sigma_2^S}(I_1^i)) \leq \Delta - 2\tau \min_{I_1^j} \Big( CBV^{\sigma_2^*}(I_1^j) - CBV^{\sigma_2^S}(I_1^j) \Big)$$  (11)

And substitute it into Equation 6,

$$\Delta + (1 - \alpha - 2\alpha\tau) \min_{I_1^j} \Big( CBV^{\sigma_2^*}(I_1^j) - CBV^{\sigma_2^S}(I_1^j) \Big)$$

$$\geq (1 - \alpha) \min_{I_1^j} \Big( v_1^\sigma(I_1^j) - CBV^{\sigma_2^S}(I_1^j) \Big) + \alpha(\Delta - 2\tau) \min_{I_1^j} \Big( CBV^{\sigma_2^*}(I_1^j) - CBV^{\sigma_2^S}(I_1^j) \Big)$$  (12)

$$\geq -\Delta$$

so that

$$CBV^{\sigma_2^S}(I_1^j) \leq CBV^{\sigma_2^*}(I_1^j) + \frac{2}{1 - (2\tau + 1)\alpha} \Delta$$  (13)

for all $I_1^j$ in the subgame.

And for infoset $I$ out of the subgame with level 0, since the refined strategy $\sigma_2^S$ and blueprint strategy $\sigma_2$ are the same here, the $CBV$ value is exactly the same and the inequality holds.

### Inductive Step

The inductive step mostly follows that of Brown and Sandholm [2017].

Since $CBV^{\sigma_2^S}(I_1) \leq CBV^{\sigma_2^*}(I_1) + \frac{2}{1-(2\tau+1)\alpha} \Delta$ holds for every subgame $S$, $\sigma_2'$ will also satisfy this inequation since $\Delta$ and $\tau$ are defined as maximum over all subgames.

Now, suppose $CBV^{\sigma_2'}(I_1) \leq CBV^{\sigma_2^*}(I_1) + \frac{2}{1-(2\tau+1)\alpha} \Delta$ holds for any infoset with level lower or equal to $k$, we will prove that it also holds for infoset with level $k + 1$.

By definition of $CBV(I_1)$,

$$CBV^{\sigma_2}(I_1, a) = \Big( \sum_{h \in I_1} \pi_{-1}^{\sigma_2}(h) v^{\langle CBR(\sigma_2), \sigma_2 \rangle}(h \cdot a) \Big) / \sum_{h \in I_1} \pi_{-1}^{\sigma_2}(h)$$

$$= \Big( \sum_{h \in I_1} \pi_{-1}^{\sigma_2}(h) \sum_{h' \in succ(h,a)} \pi_{-1}^{\sigma_2}(h, h') v^{\langle CBR(\sigma_2), \sigma_2 \rangle}(h') \Big) / \sum_{h \in I_1} \pi_{-1}^{\sigma_2}(h) \quad (14)$$

$$= \Big( \sum_{h \in I_1} \sum_{h' \in succ(h,a)} \pi_{-1}^{\sigma_2}(h') v^{\langle CBR(\sigma_2), \sigma_2 \rangle}(h') \Big) / \sum_{h \in I_1} \pi_{-1}^{\sigma_2}(h)$$

We can swap the two summations above since the game is perfect recall, then

$$CBV^{\sigma_2}(I_1, a) = \Big( \sum_{I_1' \in succ(I_1,a)} \sum_{h' \in I_1'} \pi_{-1}^{\sigma_2}(h') v^{\langle CBR(\sigma_2), \sigma_2 \rangle}(h') \Big) / \sum_{h \in I_1} \pi_{-1}^{\sigma_2}(h) \quad (15)$$

By substituting the definition of $CBV(I_1')$ into the equation above,

$$CBV^{\sigma_2}(I_1, a) = \Big( \sum_{I_1' \in succ(I_1,a)} CBV^{\sigma_2}(I_1') \sum_{h' \in I_1'} \pi_{-1}^{\sigma_2}(h') \Big) / \sum_{h \in I_1} \pi_{-1}^{\sigma_2}(h) \quad (16)$$

And by the induction hypothesis,

$$CBV^{\sigma_2}(I_1, a) \le \Big( \sum_{I_1' \in succ(I_1,a)} (CBV^{\sigma_2^*}(I_1') + \frac{2-\alpha}{1-\alpha} \Delta) \sum_{h' \in I_1'} \pi_{-1}^{\sigma_2}(h') \Big) / \sum_{h \in I_1} \pi_{-1}^{\sigma_2}(h) \quad (17)$$

Because $I_1$ is out of the subgame and $\sigma_2^*, \sigma_2$ is exactly the same outside the subgame, we will get

$$CBV^{\sigma_2}(I_1, a) \le \Big( \sum_{I_1' \in succ(I_1,a)} (CBV^{\sigma_2^*}(I_1') + \frac{2-\alpha}{1-\alpha} \Delta) \sum_{h' \in I_1'} \pi_{-1}^{\sigma_2^*}(h') \Big) / \sum_{h \in I_1} \pi_{-1}^{\sigma_2^*}(h)$$

$$= CBV^{\sigma_2^*}(I_1, a) + \frac{2-\alpha}{1-\alpha} \Delta \Big( \sum_{I_1' \in succ(I_1,a)} \sum_{h' \in I_1'} \pi_{-1}^{\sigma_2^*}(h') \Big) / \sum_{h \in I_1} \pi_{-1}^{\sigma_2^*}(h) \quad (18)$$

$$= CBV^{\sigma_2^*}(I_1, a) + \frac{2-\alpha}{1-\alpha} \Delta$$

Finally, by mathematical induction we get

$$\exp(\sigma_2') \le \exp(\sigma_2^*) + \frac{2}{1 - (2\tau+1)\alpha} \Delta \quad (19)$$

### B.2  Proof of Theorem 4.2

**Theorem 4.2** (opponent exploitation) *Let $\epsilon = \|\hat{p} - p\|_1$ be the L1 distance of the distribution $p(I_1^i)$ and $\hat{p}(I_1^i)$. Let $\eta = \min_{S \in \mathbb{S}} \max_{I_1^j \in S_{\text{top}}} \Big( CBV_1(I_1^j, \sigma_2^S) - CBV_1(I_1^j, \sigma_2^*) \Big) \ge 0$. We use $BR_p^{[\mathbb{S}, \sigma_p]}(\sigma)$ to denote the strategy for player $p$ which maximizes its utility in subgame $S \in \mathbb{S}$ against $\sigma_{-p}$ under the constraint that $BR_p^{[\mathbb{S}, \sigma_p]}(\sigma)$ and $\sigma_p$ differs only inside $\mathbb{S}$. By maximizing objective 2 for all $S \in \mathbb{S}$, the refined strategy $\sigma_2'$ satisfies*

$$u_2^{\left\langle BR_1^{[\mathbb{S}, \sigma_1]}(\sigma_2'), \sigma_2' \right\rangle}(S) \ge u_2^{\left\langle BR_1^{[\mathbb{S}, \sigma_1]}(\sigma_2^*), \sigma_2^* \right\rangle}(S) + \frac{1-\alpha}{\alpha}(\eta - 2\Delta) - \epsilon\eta \quad (20)$$

**Proof:** Still, we only consider a specific subgame $S$ first.

$\sigma_2^S$ is maximizing

$$(1-\alpha) \underbrace{\min_{I_1^j} \Big( v_1^\sigma(I_1^j) - CBV^{\sigma_2^S}(I_1^j) \Big)}_{g(\sigma_2^S)} + \alpha \underbrace{\sum_i \hat{p}(I_1^i)(v_1^\sigma(I_1^i) - CBV^{\sigma_2^S}(I_1^i))}_{f(\sigma_2^S)} \quad (21)$$

So, we have

$$(1-\alpha)g(\sigma_2^S) + \alpha f(\sigma_2^S) \geq (1-\alpha)g(\sigma_2^*) + \alpha f(\sigma_2^*) \tag{22}$$

and

$$\begin{aligned}
&\max_{I_1^j} CBV(I_1^j, \sigma_2^S) - CBV(I_1^j, \sigma_2^*) = \eta \geq \eta \\
&\Leftrightarrow g(\sigma_2^S) - \Delta \leq -\eta \\
&\Leftrightarrow g(\sigma_2^S) - \Delta \leq \Delta + g(\sigma_2^*) - \eta \quad (g(\sigma_2^*) \geq -\Delta)
\end{aligned} \tag{23}$$

Therefore,

$$\alpha f(\sigma_2^S) \geq \alpha f(\sigma_2^*) + (1-\alpha)(\eta - 2\Delta) \tag{24}$$

which means

$$\begin{aligned}
&\sum_i \hat{p}(I_1^i)(CBV^{\sigma_2^*}(I_1) - CBV^{\sigma_2^S}(I_1^i)) \\
&\geq \frac{1-\alpha}{\alpha}(\eta - 2\Delta) + \sum_i \hat{p}(I_1^i)(CBV^{\sigma_2^*}(I_1) - CBV^{\sigma_2^S}(I_1^i)) \\
&\Leftrightarrow -\sum_i \hat{p}(I_1^i)CBV^{\sigma_2^S}(I_1^i) \geq \frac{1-\alpha}{\alpha}(\eta - 2\Delta) - \sum_i \hat{p}(I_1^i)CBV^{\sigma_2^*}(I_1^i) \\
&\Leftrightarrow -\sum_i p(I_1^i)CBV^{\sigma_2^S}(I_1^i) \geq \frac{1-\alpha}{\alpha}(\eta - 2\Delta) - \sum_i p(I_1^i)CBV^{\sigma_2^*}(I_1^i) \\
&\qquad\qquad\qquad\qquad - \sum_i (p(I_1^i) - \hat{p}(I_1^i))(CBV^{\sigma_2^S}(I_1^i) - CBV^{\sigma_2^*}(I_1^i)) \\
&\Rightarrow -\sum_i p(I_1^i)CBV^{\sigma_2^S}(I_1^i) \geq \frac{1-\alpha}{\alpha}(\eta - 2\Delta) - \sum_i p(I_1^i)CBV^{\sigma_2^*}(I_1^i) - \epsilon\eta \\
&\Leftrightarrow \sum_i p(I_1^i)V_2(I_1^i, BR(\sigma_2^S), \sigma_2^S) \geq \frac{1-\alpha}{\alpha}(\eta - 2\Delta) - \epsilon\eta + \sum_i p(I_1^i)V_2(I_1^i, BR(\sigma_2^*), \sigma_2^*) \\
&\Leftrightarrow u_2^{\langle BR_1^{[\mathbb{S},\sigma_1]}(\sigma_2^{[\mathbb{S}\leftarrow\sigma_2^S]}), \sigma_2^{[\mathbb{S}\leftarrow\sigma_2^S]}\rangle}(S) \geq u_2^{\langle BR_1^{[\mathbb{S},\sigma_1]}(\sigma_2^*), \sigma_2^*\rangle}(S) + \frac{1-\alpha}{\alpha}(\eta - 2\Delta) - \epsilon\eta
\end{aligned} \tag{25}$$

Since $\eta$ is defined as minimum over all subgames, we have

$$u_2^{\langle BR_1^{[\mathbb{S},\sigma_1]}(\sigma_2'), \sigma_2'\rangle}(S) \geq u_2^{\langle BR_1^{[\mathbb{S},\sigma_1]}(\sigma_2^*), \sigma_2^*\rangle}(S) + \frac{1-\alpha}{\alpha}(\eta - 2\Delta) - \epsilon\eta \tag{26}$$

## C  Poker Rules

**Rules of Leduc Poker**

Leduc Poker is a two players game. In Leduc Poker, there are 6 cards in total, three ranks($\{J, Q, K\}$) with two suits($\{a, b\}$) each. And at the beginning, every player should put 2 chip into the pot and then will be dealt with one private card. Then, two players alternatively bet. They can call, raise and fold. If any of them fold, the game ends and all chips in the pot belongs to the other player. And when a player call, he has to put chips in the pot to ensure that he contributes equal chips as the other player in the pot. When a player raise, he has to ensure that he contributes more chips than the other player in the pot. A betting round ends when a player calls.

Leduc Poker is divided into two betting rounds. In the first round, a private card is dealt to each player and then two player start to bet. After the first betting round ends and nobody folds, a public card is dealt on board and the second betting round starts. When the second round ends, both of the player show their private hands and the stronger hands win. If a player's private card has the same rank as the public card, then he wins. Otherwise, we have $J < Q < K$ and the higher one wins. And in each betting round, there will be at most two raises in our experiment and each raise should contribute 2 more chip in the first round and 4 more chips in the second round.

Table 2: The payoff matrix of the example zero-sum game. The values are the payoffs for player 2. We will resolve for player 2.

| P2 \ P1 | L | M | R |
|---|---|---|---|
| U | 3 | 2 | 4 |
| O | 2 | 3 | 9.9 |
| D | 3 | 2 | 9.9 |
| F | -100 | -100 | 10 |

**Rules of Flop Hold'em Poker**

The rules of Flop Hold'em Poker is similar to that of Leduc Poker. In FHP, we use the standard 52-card deck. At the beginning, the first player will contribute 1 chip to the pot and the second player will contribute 2 chips. And then they will be dealt with 2 private cards each and the first player start to bet. There are still two betting rounds and the raise sizes are both 2 chips. At the end of the first betting round, there will be 3 public cards dealt on board. And the players will show their private card at the end of the second betting round and the larger one wins the game. In FHP, we have the same rule of card order as a standard Texas hold 'em .

## D    Matrix Game

In this part, we offer a matrix game as an example to show the necessity of considering safety and expected payoff simultaneously, and to demonstrate the superiority of SES over a simple mixing strategy, which follows a best response to the estimated opponent model with probability $\alpha$ and follows the blueprint with probability $1 - \alpha$.

In the matrix game shown in Table 2, let's consider two specific NEs. In both NEs, P1 will play L/M with $0.5$ probability each. P2 will play U/O with $0.5$ probability in the first NE and O/D with $0.5$ probability in the second NE. Suppose the blueprint strategy is the first NE. Consider the case when P1 plays a rather weak strategy that he will only play R. We apply SES to search for P2's refined strategy.

When the estimation of opponent strategy is accurate such that $\hat{p} = p$, the best response of P2 is always playing F, which is highly exploitable, while SES finds the second NE under proper $\alpha$. To give more details, the exploitability and expected payoff of the strategy found by SES and the mixing strategy are shown in Figure 4. We can see that SES achieves both lower exploitability and better performance than the mixing strategy at almost all $\alpha$ values.

Theoretically, for a mixing strategy $\text{mix}_p(\sigma_1, \sigma_1')$ which plays $\sigma_1$ with probability $p$, and $\sigma_1'$ with $1 - p$ [Johanson et al., 2007], we have the following propositions.

**Proposition D.1.** *Utility of a mixing strategy:* $u(\text{mix}_p(\sigma_1, \sigma_1'), \sigma_2) = pu(\sigma_1, \sigma_2) + (1-p)u(\sigma_1', \sigma_2)$.

**Proposition D.2.** *Exploitability of a mixing strategy:* $\exp(\text{mix}_p(\sigma_1, \sigma_1')) \leq p \exp(\sigma_1) + (1 - p) \exp(\sigma_1')$. *The example in Figure 4 shows that the bound is tight.*

Compared with the corresponding theorems 4.1 and 4.2 of SES, we find that the simple mixing strategy cannot provide a safety guarantee and is inferior to SES. In some circumstances, $\exp(\sigma_{\text{unsafe}})$ can be quite large. Therefore, the exploitability of the mixing strategy can also be large even for a rather small $p$. In contrast, as shown in Theorem 1, SES's exploitability does not rely on $\exp(\sigma_{\text{unsafe}})$, which makes our bound for SES tighter than that of mixed strategy in most cases.

## E    Implementation Details

**Leduc Poker.** In Leduc Poker, we solve for a blueprint strategy using a variant of CFR algorithm [Lanctot et al., 2009, Tammelin et al., 2015] with 1M iterations in the full game. Then we apply search in subgames when the board card is dealt.

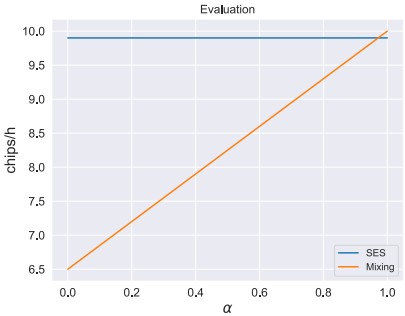 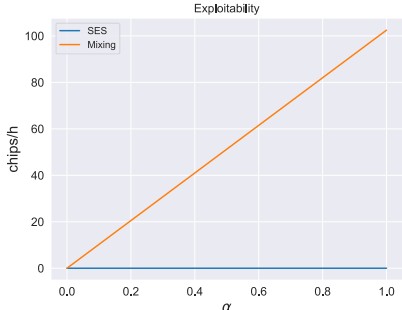

Figure 4: **Left:** Expected payoffs of SES and the mixing strategy in the proposed matrix game example. **Right:** Exploitability of the two algorithms.

**Flop Hold'em Poker (FHP).** As for FHP, there are 1,286,792 different infosets for each betting sequence. We cluster them into 200 infosets by an abstraction algorithm [Johanson et al., 2013] in order to make equilibrium finding feasible. Then, we compute a blueprint strategy in this abstraction with 10,000,000 iterations. We apply search immediately once the flop cards are dealt.

**Generate estimated opponent strategy.** Given $p$ and $\epsilon$, we use the following procedure to generate the corresponding $\hat{p}$ such that $|p - \hat{p}| = \epsilon$. 1. In the beginning, we set $\hat{p} = p$. 2. We randomly choose a subset $S_+$ from index set $S$ of $\hat{p}$ and we set $S_- = S - S_+$. We ensure that $\sum_{i \in S_+}(1.0 - \hat{p}(i)) > \epsilon/2$ and $\sum_{i \in S_-} \hat{p}(i) > \epsilon/2$ by the following procedure: to get $S_+, S_-$, we will randomly permute the index of $\hat{p}$ first to get permutation $A$. Then, we will sample $k \sim Uniform(1, |A|)$. We will check whether $S_+ = \{A_1, ..., A_k\}$ and $S_- = \{A_{k+1}, ..., A_{|A|}\}$ satisfy the condition above. We will continue the process until $S_+, S_-$ satisfy the condition above. 3. We divide $\epsilon/2$ into 10,000 pieces with value $\frac{\epsilon}{20,000}$ assigned to each piece. 4. For $S_+$, we will do reject sampling to sample 10,000 pieces to different indexes. We will pick $i \in S_+$ randomly each time. If $\hat{p}(i) + \frac{\epsilon}{20,000} < 1.0$, we will assign one piece to $i$ and increase $\hat{p}(i)$ by $\frac{\epsilon}{20,000}$. The process will continue until we have assigned all $10,000$ pieces. We will do the same thing for $S_-$ except that only when $p(i) - \frac{\epsilon}{20,000} > 0$ we will assign one piece to $i$.

# F  Comparison with EXP-STRATEGY

Table 3, 4, and 5 show the comparison between SES and EXP-STRATEGY.

# G  Theoretical Bound of EXP-STRATEGY

Similar to SES, EXP-STRATEGY maximizes the following objective,

$$(1 - \alpha) \min_{I_1^j} \left( v_1^\sigma(I_1^j) - CBV^{\sigma_2^S}(I_1^j) \right) + \alpha \sum_i \hat{p}(I_1^i)(v_1^\sigma(I_1^i) - v_1^{\langle \hat{\sigma}_1, \sigma_2^S \rangle}(I_1^i)) \qquad (27)$$

**Theorem G.1.** *(safety) Let $\mathbb{S}$ be a disjoint set of subgames $S$. Let $\sigma^* = \langle \sigma_1^*, \sigma_2^* \rangle$ be the NE where P2's strategy is constrained to be the same with $\sigma_2$ outside $\mathbb{S}$. Define $\Delta = \max_{S \in \mathbb{S}, I_1^i \in S_{top}} |CBV_1^{\sigma_2^*}(I_1^i) - v_1^\sigma(I_1^i)|$. We use $BR_p^{[\mathbb{S}, \sigma_p]}(\sigma)$ to denote the strategy for player $p$ which maximizes its utility in subgame $S \in \mathbb{S}$ against $\sigma_{-p}$ under the constraint that $BR_p^{[\mathbb{S}, \sigma_p]}(\sigma)$ and $\sigma_p$ differs only inside $\mathbb{S}$. Let $\hat{p}(I_1^i)$ be the estimation of reach probability $p(I_1^i)$ given by the real opponent strategy. Let $\epsilon = \|\hat{p} - p\|_1$ be the L1 distance of the distribution $p(I_1^i)$ and $\hat{p}(I_1^i)$. Define $\Omega = \max_S \max_i (CBV^{\sigma_2^*}(I_1^i) + CBV^{\hat{\sigma}_1}(I_1^i))$. We have a bounded exploitability given by:*

$$\exp(\sigma_2') \le \exp(\sigma_2^*) + 2\Delta + \frac{\alpha}{1-\alpha}\left( u_1^{\left\langle BR_1^{[\mathbb{S}, \sigma_1]}(\sigma_2^*), \sigma_2^* \right\rangle}(S) + u_2^{\left\langle BR_2^{[\mathbb{S}, \sigma_2]}(\hat{\sigma}_1), \hat{\sigma}_1 \right\rangle}(S) \right) + \frac{\epsilon}{1-\alpha}\Omega$$
$$(28)$$

Table 3: Exploitability and evaluation performance of SES and EXP-STRATEGY, when strategy estimation error is 0.1.

| $\text{Pr}_{shuffle}$ | $\alpha$ | EXPLOITABILITY | | EVALUATION | |
|---|---|---|---|---|---|
| | | EXP-STRATEGY | SES | EXP-STRATEGY | SES |
| 0.0 | 0.0 | 29.43($\pm$0.01) | 29.43($\pm$0.01) | 1.20($\pm$0.01) | 1.20($\pm$0.01) |
| | 0.3 | 34.64($\pm$0.11) | 28.01($\pm$0.07) | -2.55($\pm$0.07) | 2.28($\pm$0.03) |
| | 0.7 | 132.06($\pm$0.68) | 41.59($\pm$0.37) | -34.75($\pm$0.14) | 1.48($\pm$0.08) |
| | 1.0 | 969.77($\pm$6.92) | 55.54($\pm$1.08) | -108.02($\pm$0.93) | -1.97($\pm$0.23) |
| 0.3 | 0.0 | 29.43($\pm$0.01) | 29.43($\pm$0.00) | 5.15($\pm$0.19) | 5.15($\pm$0.19) |
| | 0.3 | 34.75($\pm$0.13) | 28.06($\pm$0.04) | 1.63($\pm$0.40) | 6.21($\pm$0.25) |
| | 0.7 | 134.37($\pm$2.59) | 41.95($\pm$0.39) | -27.22($\pm$0.91) | 6.00($\pm$0.81) |
| | 1.0 | 981.47($\pm$11.86) | 55.10($\pm$1.13) | -88.94($\pm$6.07) | 3.30($\pm$0.96) |
| 0.7 | 0.0 | 29.43($\pm$0.01) | 29.43($\pm$0.01) | 12.35($\pm$1.52) | 12.34($\pm$1.52) |
| | 0.3 | 34.84($\pm$0.12) | 28.10($\pm$0.02) | 9.68($\pm$1.39) | 13.55($\pm$1.58) |
| | 0.7 | 137.72($\pm$2.55) | 41.90($\pm$0.13) | -13.80($\pm$2.08) | 13.96($\pm$1.42) |
| | 1.0 | 985.82($\pm$30.43) | 55.79($\pm$0.50) | -61.84($\pm$5.38) | 11.61($\pm$1.56) |

Table 4: Exploitability and evaluation performance of SES and EXP-STRATEGY, when strategy estimation error is 0.3.

| $\text{Pr}_{shuffle}$ | $\alpha$ | EXPLOITABILITY | | EVALUATION | |
|---|---|---|---|---|---|
| | | EXP-STRATEGY | SES | EXP-STRATEGY | SES |
| 0.0 | 0.0 | 29.43($\pm$0.01) | 29.43($\pm$0.01) | 1.20($\pm$0.01) | 1.19($\pm$0.03) |
| | 0.3 | 31.49($\pm$0.27) | 26.77($\pm$0.07) | 3.19($\pm$0.06) | 2.54($\pm$0.04) |
| | 0.7 | 57.80($\pm$2.70) | 26.18($\pm$0.67) | 0.81($\pm$0.53) | 3.68($\pm$0.06) |
| | 1.0 | 589.02($\pm$9.71) | 30.14($\pm$2.51) | -52.78($\pm$1.76) | 4.36($\pm$0.23) |
| 0.3 | 0.0 | 29.43($\pm$0.01) | 29.43($\pm$0.00) | 5.15($\pm$0.19) | 5.15($\pm$0.19) |
| | 0.3 | 31.61($\pm$0.25) | 26.80($\pm$0.03) | 8.40($\pm$0.87) | 6.45($\pm$0.24) |
| | 0.7 | 64.77($\pm$4.49) | 26.44($\pm$0.53) | 11.31($\pm$2.90) | 7.69($\pm$0.38) |
| | 1.0 | 618.06($\pm$16.53) | 30.04($\pm$1.17) | -25.57($\pm$2.38) | 8.60($\pm$0.53) |
| 0.7 | 0.0 | 29.43($\pm$0.01) | 29.43($\pm$0.01) | 12.35($\pm$1.52) | 12.35($\pm$1.52) |
| | 0.3 | 32.20($\pm$0.33) | 26.85($\pm$0.02) | 17.12($\pm$0.97) | 13.73($\pm$1.56) |
| | 0.7 | 73.91($\pm$1.56) | 27.05($\pm$0.35) | 28.26($\pm$0.30) | 15.25($\pm$1.44) |
| | 1.0 | 600.82($\pm$5.67) | 32.48($\pm$1.41) | 15.38($\pm$4.74) | 16.46($\pm$1.31) |

**Proof:** For any subgame $S \in \mathbb{S}$, we have

$$(1-\alpha)\Delta + (1-\alpha)\min_{I_1^j}(CBV^{\sigma_2^*}(I_1^j) - CBV^{\sigma_2^S}(I_1^j)) + \alpha \sum_i p(I_1^i)CBV^{\hat{\sigma}_1}(I_1^i) + \epsilon \max_i CBV^{\hat{\sigma}_1}(I_1^i)$$

$$\geq (1-\alpha)\min_{I_1^j}(v_1^\sigma(I_1^j) - CBV^{\sigma_2^S}(I_1^j)) + \alpha \sum_i p(I_1^i)CBV^{\hat{\sigma}_1}(I_1^i) + \epsilon \max_i CBV^{\hat{\sigma}_1}(I_1^i)$$

$$\geq (1-\alpha)\min_{I_1^j}(v_1^\sigma(I_1^j) - CBV^{\sigma_2^S}(I_1^j)) + \alpha \sum_i \hat{p}(I_1^i)CBV^{\hat{\sigma}_1}(I_1^i)$$

$$\geq (1-\alpha)\min_{I_1^j}(v_1^\sigma(I_1^j) - CBV^{\sigma_2^S}(I_1^j)) + \alpha \sum_i \hat{p}(I_1^i)(-v_1^{\langle \hat{\sigma}_1, \sigma_2^S \rangle}(I_1^i))$$

$$\geq (1-\alpha)\min_{I_1^j}(v_1^\sigma(I_1^j) - CBV^{\sigma_2^*}(I_1^j)) + \alpha \sum_i \hat{p}(I_1^i)(-v_1^{\langle \hat{\sigma}_1, \sigma_2^* \rangle}(I_1^i))$$

$$\geq -(1-\alpha)\Delta - \alpha \sum_i \hat{p}(I_1^i)CBV^{\sigma_2^*}(I_1^i)$$

$$\geq -(1-\alpha)\Delta - \alpha \sum_i p(I_1^i)CBV^{\sigma_2^*}(I_1^i) - \epsilon \max_i CBV^{\sigma_2^*}(I_1^i)$$

$$(29)$$

Table 5: Exploitability and evaluation performance of SES and EXP-STRATEGY, when strategy estimation error is 0.5.

| $\Pr_{shuffle}$ | $\alpha$ | EXPLOITABILITY | | EVALUATION | |
| | | EXP-STRATEGY | SES | EXP-STRATEGY | SES |
|---|---|---|---|---|---|
| 0.0 | 0.0 | 29.43($\pm$0.00) | 29.43($\pm$0.01) | 1.20($\pm$0.01) | 1.20($\pm$0.01) |
| | 0.3 | 34.76($\pm$0.21) | 28.01($\pm$0.07) | -2.32($\pm$0.11) | 2.28($\pm$0.03) |
| | 0.7 | 156.79($\pm$3.54) | 41.59($\pm$0.37) | -39.72($\pm$0.70) | 1.48($\pm$0.08) |
| | 1.0 | 1020.47($\pm$10.09) | 55.54($\pm$1.08) | -114.78($\pm$0.47) | -1.97($\pm$0.23) |
| 0.3 | 0.0 | 29.43($\pm$0.01) | 29.43($\pm$0.01) | 5.15($\pm$0.19) | 5.15($\pm$0.19) |
| | 0.3 | 34.90($\pm$0.21) | 28.06($\pm$0.04) | 1.85($\pm$0.51) | 6.21($\pm$0.25) |
| | 0.7 | 154.67($\pm$8.22) | 41.95($\pm$0.39) | -31.78($\pm$1.64) | 6.00($\pm$0.81) |
| | 1.0 | 1024.30($\pm$15.31) | 55.10($\pm$1.13) | -96.89($\pm$5.96) | 3.30($\pm$0.96) |
| 0.7 | 0.0 | 29.43($\pm$0.01) | 29.43($\pm$0.01) | 12.34($\pm$1.52) | 12.34($\pm$1.52) |
| | 0.3 | 34.95($\pm$0.34) | 28.10($\pm$0.02) | 9.58($\pm$1.29) | 13.55($\pm$1.58) |
| | 0.7 | 158.98($\pm$5.16) | 41.90($\pm$0.13) | -17.76($\pm$1.96) | 13.96($\pm$1.42) |
| | 1.0 | 1019.04($\pm$28.61) | 55.79($\pm$0.50) | -70.64($\pm$4.37) | 11.61($\pm$1.56) |

where the fourth line to the fifth line is derived from the fact that the fourth line is exactly the objective of $\sigma_2^S$ subtracted a constant .

That is,

$$
\begin{aligned}
\min_{I_1^j}(CBV^{\sigma_2^*}(I_1^j) - CBV^{\sigma_2^S}(I_1^j)) \geq &-2\Delta - \frac{\alpha}{1-\alpha}\sum_i p(I_1^i)\Big(CBV^{\sigma_2^*}(I_1^i) \\
&+ CBV^{\hat{\sigma}_1}(I_1^i)\Big) - \frac{\epsilon}{1-\alpha}(\max_i CBV^{\sigma_2^*}(I_1^i) + \max_i CBV^{\hat{\sigma}_1}(I_1^i))
\end{aligned}
\tag{30}
$$

Follows the same induction step,

$$
\exp(\sigma_2^S) \leq \exp(\sigma_2^*) + 2\Delta + \frac{\alpha}{1-\alpha}\Big(u_1^{\left\langle BR_1^{[\mathbb{S},\sigma_1]}(\sigma_2^*),\sigma^* \right\rangle}(S) + u_2^{\left\langle BR_2^{[\mathbb{S},\sigma_2]}(\hat{\sigma}_1),\hat{\sigma}_1 \right\rangle}(S)\Big) + \frac{\epsilon}{1-\alpha}\Omega
\tag{31}
$$

Therefore, $\sigma_2'$ also satisfies the inequality above since every subgame of it satisfies that.

**Theorem G.2.** *(opponent exploitation) Let $\epsilon = \|\hat{p} - p\|_1$ be the L1 distance of the distribution $p(I_1^i)$ and $\hat{p}(I_1^i)$. Let $\eta = \min_{S \in \mathbb{S}} \max_{I_1^j \in S_{\text{top}}} \Big(CBV_1(I_1^j, \sigma_2^S) - CBV_1(I_1^j, \sigma_2^*)\Big) \geq 0$ and $\Delta = \max_{S \in \mathbb{S}, I_1^i \in S_{top}} |CBV_1^{\sigma_2^*}(I_1^i) - v_1^\sigma(I_1^i)|$. By maximizing objective [27], for all $S \in \mathbb{S}$, the refined strategy $\sigma_2'$ satisfies*

$$
u_2^{\langle \hat{\sigma}_1, \sigma_2' \rangle}(S) \geq u_2^{\langle \hat{\sigma}_1, \sigma_2^* \rangle}(S) + \frac{1-\alpha}{\alpha}(\eta - 2\Delta) - \epsilon\Omega
\tag{32}
$$

**Proof:**

Still, we only consider a specific subgame $S$ first.

$\sigma_2^S$ is maximizing

$$
(1-\alpha)\underbrace{\min_{I_1^j}\Big(v_1^\sigma(I_1^j) - CBV^{\sigma_2^S}(I_1^j)\Big)}_{g(\sigma_2^S)} + \alpha\underbrace{\sum_i \hat{p}(I_1^i)(v_1^\sigma(I_1^i) - v_1^{\langle \hat{\sigma}_1, \sigma_2^S \rangle}(I_1^i))}_{f(\sigma_2^S)}
\tag{33}
$$

So, we have

$$
(1-\alpha)g(\sigma_2^S) + \alpha f(\sigma_2^S) \geq (1-\alpha)g(\sigma_2^*) + \alpha f(\sigma_2^*)
\tag{34}
$$

and

$$\max_{I_1^j} CBV(I_1^j, \sigma_2^S) - CBV(I_1^j, \sigma_2^*) \geq \eta$$

$$\Leftrightarrow g(\sigma_2^S) - \Delta \leq -\eta \tag{35}$$

$$\Leftrightarrow g(\sigma_2^S) - \Delta \leq \Delta + g(\sigma_2^*) - \eta \quad (g(\sigma_2^*) \geq -\Delta)$$

Therefore,

$$\alpha f(\sigma_2^S) \geq \alpha f(\sigma_2^*) + (1 - \alpha)(\eta - 2\Delta) \tag{36}$$

which means

$$\sum_i \hat{p}(I_1^i)(v_1^{\langle \hat{\sigma}_1, \sigma_2^* \rangle}(I_1^i) - v_1^{\langle \hat{\sigma}_1, \sigma_2^S \rangle}(I_1^i)) \geq \frac{1-\alpha}{\alpha}(\eta - 2\Delta)$$

$$\Leftrightarrow -\sum_i \hat{p}(I_1^i)v_1^{\langle \hat{\sigma}_1, \sigma_2^S \rangle}(I_1^i) \geq \frac{1-\alpha}{\alpha}(\eta - 2\Delta) - \sum_i \hat{p}(I_1^i)v_1^{\langle \hat{\sigma}_1, \sigma_2^* \rangle}(I_1^i)$$

$$\Leftrightarrow -\sum_i p(I_1^i)v_1^{\langle \hat{\sigma}_1, \sigma_2^S \rangle}(I_1^i) \geq \frac{1-\alpha}{\alpha}(\eta - 2\Delta) - \sum_i p(I_1^i)v_1^{\langle \hat{\sigma}_1, \sigma_2^* \rangle}(I_1^i)$$

$$\qquad\qquad - \sum_i (p(I_1^i) - \hat{p}(I_1^i))(v_1^{\langle \hat{\sigma}_1, \sigma_2^S \rangle}(I_1^i) - v_1^{\langle \hat{\sigma}_1, \sigma_2^* \rangle}(I_1^i))$$

$$\Rightarrow -\sum_i p(I_1^i)v_1^{\langle \hat{\sigma}_1, \sigma_2^S \rangle}(I_1^i) \geq \frac{1-\alpha}{\alpha}(\eta - 2\Delta) - \sum_i p(I_1^i)v_1^{\langle \hat{\sigma}_1, \sigma_2^* \rangle}(I_1^i) - \epsilon(\max CBV^{\sigma_2^*}(I_1^i) + \max CBV^{\hat{\sigma}_1}(I_1^i))$$

$$\Leftrightarrow \sum_i p(I_1^i)v_2^{\langle \hat{\sigma}_1, \sigma_2^S \rangle}(I_1^i) \geq \frac{1-\alpha}{\alpha}(\eta - 2\Delta) - \epsilon(\max CBV^{\sigma_2^*}(I_1^i) + \max CBV^{\hat{\sigma}_1}(I_1^i)) + \sum_i p(I_1^i)v_2^{\langle \hat{\sigma}_1, \sigma_2^* \rangle}(I_1^i) \tag{37}$$

Since $\eta$ is defined as minimum over all subgames and $\Omega$ is defined as maximum over all subgames, we have

$$u_2^{\langle \hat{\sigma}_1, \sigma_2' \rangle}(S) \geq u_2^{\langle \hat{\sigma}_1, \sigma_2^* \rangle}(S) + \frac{1-\alpha}{\alpha}(\eta - 2\Delta) - \epsilon\Omega \tag{38}$$

## H   Choosing $\alpha$

We find that automatically choosing $\alpha$ by explicitly incorporating modeling accuracy into search formulation is non-trivial. It still requires additional information like the risk you are willing to take.

(i) Without further specifications, there is always a trade-off between safety and exploitation regardless of opponent modeling accuracy. Since the prediction is made based on past observation, high accuracy (confidence) cannot guarantee the success in the future (since we do not have the assumption that the opponent will not change his behavior in the next round). Therefore, to incorporate accuracy(confidence) into objective, we need more information like the risk you are willing to take. Quantifying the relationship between modeling confidence and $\alpha$ theoretically will require more assumptions about the opponent's behavior than this paper does.

(ii) Empirically, exploitation and safety correspond to a fixed $\alpha$ and prediction accuracy differ in different games (FHP and Leduc) as our experiments imply.

Practically, we can use learning algorithms (e.g., bandit) to automatically select $\alpha$ in each round with respect to one's specific goal.

## I   Potential Negative Social Impacts

Malicious uses of algorithmic exploitation techniques may cause negative social impacts. For instance, in online advertising or e-commerce, improper usage of irrationality exploitation algorithms may lure people into a scam. In some social areas, algorithmic exploitation may exacerbate inequality and cause extra harm to vulnerable groups. The method that we propose is general-purpose, and

positive/negative social impacts are generally related to specific applications. For instance, in training or education, exploitation algorithms can help trainees or students to discover their shortcomings and have opportunities to improve. Since the exploitation algorithm requires human data to build an agent model and estimate the distribution $\hat{p}$, actions can be taken from the privacy data protection side to prevent such malicious abuse. We believe that regulations and laws should be established to restrain any unethical applications and protect human privacy data.