# OpenReview forum: "Safe Opponent-Exploitation Subgame Refinement"
_NeurIPS.cc/2022/Conference — NeurIPS 2022 Accept_

### Official Review · Reviewer_cYqF · 2022-07-11

**Rating:** 6
**Confidence:** 4
**Soundness:** 4 excellent
**Presentation:** 3 good
**Contribution:** 3 good

**Summary:**

In zero-sum games, opponents with limited rationality can be exploited if they do not play Nash equilibrium strategy.
An exploiter can refine its precomputed blueprint (an approximate NE strategy) in the visited subgames as the game is played online, to take advantage of the opponent's irrational play, using Safe Exploitation Search (SES).
The refinement is done by optimizing a function which trades off safety and exploitation, set by a hyperparameter "alpha".
Authors show safety and opponent exploitation bounds based on this hyperparameter, and uncertainty in modelling the opponent and approximation error of the blueprint or newly found strategy.
In practical terms, the player's strategy in a subgame is found using a special gadget game, which is an alpha-combination of max-margin gadget (for safety) and value solving subgame (for opponent exploitation).

**Questions:**

What is the reason for using precomputed blueprint for P2? Would not the algorithm work with no blueprint strategy? If I understand it correctly, blueprint is used only for theoretical derivations, to ensure strategy is well defined for the whole game (outside of the searched path).


**Limitations:**

Author claim their work does not pose negative societal impacts. Although opponent exploitation in games like Poker can have limited social impact, in general algorithmic exploitation of human irrationality in decision-making under uncertainty can have grave negative social impact and should be properly discussed.


**Strengths And Weaknesses:**


Strengths:

- Explicit bounds on safety and opponent exploitation in terms of alpha.
- Empirical results on two poker games that show safety and exploitation as function of tradeoff alpha, and with different opponent modelling errors.
- Comparison with prior algorithm RNR, which is more computationally demanding. SES significantly outperforms RNR in both safety and exploitation.
- Ablation study of SES, where opponent strategy is fixed in the value-solving subgame -- performs worse than original. (A possible takeaway is that ranges+CBVs are enough for exploitation search).

Weaknesses:

- Authors do not explain how they compute opponent CBVs needed for max-margin gadget. (I suspect they computed these recursively over the whole game tree).
  Computing opponent CBVs can be non-trivial in large-scale problems, especially when we try to incoporate opponent modelling. This relates to following point: The work does not use depth-limited subgames (with value function evaluations), and cannot be therefore used efficiently to solve large-scale problems. In fact, authors claim L50 "enables our method’s scalability to large games such as Texas Hold’em", which they i) do not demonstrate and ii) is probably difficult due to computation of opponent CBVs.
  I advise this to be mentioned explicitly in the paper and weaken the claim of scalability. (i.e. claim it is future work, and not done by this paper).
- L7-8: "SES smoothly interpolates between the two extremes of online strategy refinement" -- There is no "smoothness" proposition in the paper, only  upper and lower bounds on exploitability and safety, respectively. (For a claim of smoothness I would expect something like a proof of Lipschitz continuity with respect to hyperparameter alpha.)


Other comments:

- L246-L247: Since σ^S_1 is the best-response to σ^S_2 and the two parts only differ at how to go to each infoset of player 1, player 1 will also keep his strategy the same in both parts. -- This is likely not true. If it was, there is no point in creating copies of the S into S_1 and S_2 in the first place? The top chance node/P1 node can change the expected utilities u(z)*reach_prob_{1,2,c}(z) and hence PL1 NE strategy in the two subgames.
- L248: I^i_1 unclear which infosets these refer to, are these the top infosets of the subgame S?
- L249: "the objective of a Nash Equilibrium of p2" -- NE are solutions to the max-min optimization. Just maximizing for one player is not enough to find a NE in a game.
- L250: Maximization is not well defined.
- Appendix A: Resolve Current Node is not defined.
- Appendix D: Figure 4: Top/Bottom should be Left/Right.
- The $\mathcal{S}$ is used as a disjoint set of subgames S (Thm 4.1), but should it not be the set set of subgames visited in an online play? (I.e from root to terminal).

---

> ### Author Response · Authors · 2022-07-30
> **Response to reviewer cYqF (2/2)**
>
> 10. **Q**: Usage of the blueprint of P2.
>
>     **A**: As explained in Question 1, we use the blueprint of P2 in the resolving process. In Eq (2), the term $v_1^{\sigma}(I_1^j)$ is computed by running the blueprint in the subgame, which requires both the blueprint of player 1 and player 2.
>
> 11. **Q**: Negative social impact of general algorithmic exploitation of human irrationality in decision-making.
>
>     **A**: Thank you for your suggestion. Malicious uses of algorithmic exploitation techniques may cause negative social impacts. For instance, in online advertising or e-commerce, improper usage of irrationality exploitation algorithms may lure people into a scam. In some social areas, algorithmic exploitation may exacerbate inequality and cause extra harm to vulnerable groups.
>
>     The method that we propose is general-purpose, and positive/negative social impacts are generally related to specific applications. For instance, in training or education, exploitation algorithms can help trainees or students to discover their shortcomings and have opportunities to improve.
>
>     Since the exploitation algorithm requires human data to build an agent model and estimate the distribution $\hat p$, actions can be taken from the privacy data protection side to prevent such malicious abuse. We believe that regulations and laws should be established to restrain any unethical applications and protect human privacy data.

---

> ### Author Response · Authors · 2022-07-30
> **Response to reviewer cYqF (1/2)**
>
> We appreciate the reviewer for the insightful comments.
>
> 1. **Q**: How can the algorithm be extended to large-scale problems?
>
>     **A**: Thanks for the author's comments. Our algorithm can be easily extended to depth-limited search by changing S1 and S2 to subgames with function evaluations at the bottom.
>
>     We explain the technical details in the following. We run CFR in the gadget game to solve for the refined safe exploitation strategy rather than optimizing objective (2) directly. As can be seen in the objective and the gadget game construction process (section 4.2), we use $v_1^{\sigma}(I_1^j)$ to approximate $CBV_1^{\sigma_2^*}(I_1^j)$, where $\sigma$ is the pre-computed blueprint strategy. As claimed by [Brown and Sandholm, 2017] (section 6, Estimates for Alternative Payoffs), $v_1^{\sigma}(I_1^j)$ can be a good approximation for $CBV_1^{\sigma_2^*}(I_1^j)$. The reason is that $\sigma_1$ can be viewed as a best-response to $\sigma_2$ in the abstracted subgame (they are a pair of NE) and $\sigma_2$ is a NE in the abstracted game. This can be computed rather fast since it is only an evaluation in the subgame which enables Monte-Carlo approach to efficiently approximate the expected utility.
>
>
> 3. **Q**: The usage of "smoothness".
>
>     **A**: We are sorry for the confusion caused by the usage of "smoothness". Here we mean that by tuning $\alpha$ from 0 to 1, SES can vary continuously from being completely conservative (NE) to exploiting the opponent only. We have refined the expression in the latest revision.
>
> 4. **Q**: L246-L247: Is the strategy of both players in S1 and S2 the same?
>
>     **A**: Thank you for your comment and we have added more explanations in our latest version. First of all, S1 and S2 are the same for player 2 because it cannot tell which subgame it is in. Note that S1 and S2 only differ in the distribution of the infosets of player 1. Given the strategies of player 2 in S1 and S2 are the same, the difference between the distribution of infosets of player 1 on the top of the subgame will not change the counterfactual value of player 1. Therefore, we will have the same counterfactual regret and the same strategies generated by CFR for player 1 in S1 and S2.
>
>     Secondly, the reason that we still need S1 and S2 is that the distribution of S1 at the top is not known beforehand. It is fully determined by player 1 which means that we cannot use a chance node at the top of S1. Therefore, we need to split S1 and S2 so that we can have a player node at S1 and a chance node at S2.
>
> 5. **Q**: L248: Is $I_1^i$ the top infoset of the subgame $S$?
>
>     **A**: Yes, they are the infosets at the top of the subgame.
>
> 6. **Q**: Explanations for maximizing $-CBV_1^{\sigma_2^S}(I_1^i)$ and $v_1^{\sigma}(I_1^i)-CBV_1^{\sigma_2^S}(I_1^i)$.
>
>    **A**: Note that $\max_{\sigma_2^S}-CBV_1^{\sigma_2^S}(I_1^i)$ is equivalent to $\max_{\sigma_2^S}\min_{\sigma_1^S} -v_1^{\sigma_1^S, \sigma_2^S}(I_1^i)$, which is indeed a max-min optimization.
>    Since the optimization objective (2) is hard to compute directly, we construct a gadget game such that the NE in the gadget game is exactly the solution to the original optimization problem. Therefore, we can run any NE-finding algorithm in the gadget game. The step 2 in the gadget game construction (adding $v_1^{\sigma}(I_1^j)$ to $u_2(z)$ at all leaves) is to construct the term $v_1^{\sigma}(I_1^j)-CBV_1^{\sigma_2^S}(I_1^j)$ in objective (2).
>
> 7. **Q**: Appendix A: Resolve Current Node is not defined?
>
>     **A**: It is a boolean variable indicating whether the current node will do a subgame resolving.
>
> 8. **Q**: Figure 4: Top/Bottom should be Left/Right.
>
>     **A**: Thank you for your comment. We have already modified it in the latest version.
>
> 9. **Q**: Why is $\mathbb S$ defined as a disjoint set of subgames S (Thm 4.1)?
>
>    **A**: We use an example to explain why $\mathbb{S}$ is defined in this way. For instance, in FHP, we conduct a subgame resolving at the beginning of the flop round, so $\mathbb{S}$ is the set of all the subgames at the beginning of the flop round. We will always perform such resolving for any $S\in \mathbb{S}$ encountered. Therefore, our strategy is equivalent to "based on blueprint, performing subgame resolving on all $S\in \mathbb{S}$", although only one subgame is actually resolved in one round of online play. Therefore, in theoretical analysis, we care about the strategy which performs resolving in all $S\in \mathbb{S}$. The definition of $\mathbb{S}$ is widely used in previous literature (e.g., Theorem 1 in [Brown and Sandholm, 2017]).
>
>
> Brown, Noam, and Tuomas Sandholm. "Safe and nested subgame solving for imperfect-information games." Advances in neural information processing systems 30 (2017).

---

### Official Review · Reviewer_ByZa · 2022-07-12

**Rating:** 4
**Confidence:** 4
**Soundness:** 4 excellent
**Presentation:** 3 good
**Contribution:** 2 fair

**Summary:**

This paper introduces a search algorithm for imperfect-information games that balances between opponent exploitation and being unexploitable by adding a parameter to interpolate between the two extremes. The authors present experimental results in a variety of games showing that the algorithm achieves most of the benefits of both, having both low exploitability and is able to exploit effectively.

**Questions:**

Are there differences between this algorithm and the one used in DeepStack?

**Limitations:**

The authors have adequately addressed limitations.

**Strengths And Weaknesses:**

The paper is well-written and easy to follow. It places its results well in the context of related search algorithms. Also, the results are convincing.

The weakness is the novelty. The algorithm essentially just interpolates between two already-known search algorithms. Moreover, I think the algorithm is essentially identical to the one already used in DeepStack (see the "Opponent Ranges in Resolving" section in the appendix). The main difference being that the authors propose using a model of the opponent as the policy, whereas DeepStack used its own policy net.

Update: Based on the author responses it's true that DeepStack is not the right comparison. However, the technique in the paper does appear to be a rather straightforward interpolation of Maxmargin and Unsafe search. For that reason I am still somewhat concerned about the novelty of the algorithm.

---

> ### Author Response · Authors · 2022-07-30
> **Response to reviewer ByZa**
>
> We thank the reviewer for recognizing that the paper is well-written and provides convincing results. However, we respectfully disagree that our algorithm is the same as that in DeepStack. We aim to address a problem quite different from that in DeepStack, which is well-motivated (supported by reviewer Zzn9). Our method is novel in the following perspectives.
>
> 1. **Q**: Difference from Deepstack?
>
>    **A**: We have provided a short discussion about DeepStack in section 2 (line 107). The detailed differences are mainly three-fold.
>
>    (i.) Our motivation is different from that of DeepStack. In DeepStack, they used a mixing strategy to accelerate the subgame resolving process and we used it to exploit the weakness of the opponent.
>
>    (ii.) The technical details are different. DeepStack gave two subgame resolving variants (see opponent ranges in re-solving in the appendix). One is to mix the estimation of opponent ranges $\hat p$ with a uniform private hand distribution and the opponent can choose whether to terminate or not after receiving her private card (which we refer to as Method 1 in the following). The second is that the opponent has $1-\alpha$ probability to get into a subgame with a **uniform** private hand distribution of herself where she can choose whether to terminate or not after receiving her private card, and $\alpha$ probability to get into a subgame with an estimated distribution of private hand where she **cannot** determine whether to terminate or not (which we refer to as Method 2 in the following).
>
>    Method 1 in DeepStack is quite different from ours. Firstly, Method 1 does not split the subgame into S1 and S2 as our approach does (Figure 1). It simply changes the distribution of the subgame into a mixture of the estimation of the opponent and a uniform strategy. Therefore, as a result, Method 1 cannot exploit the weakness of the opponent since the opponent will simply terminate the game when its private hands are not satisfactory.
>
>    Additionally, contrary to Method 2 in DeepStack, our approach allows the opponent to get into a subgame with $1-\alpha$ probability, where the opponent can choose her private hand **arbitrarily**. In this way, our approach can get a tighter bound on safety (in contrast, Method 2 in DeepStack did not have any bound). The reason is that we solve for the strategy when the opponent can choose her own best private card, which would certainly be stronger than receiving a private card from a uniform distribution, and determine whether to terminate to get a fixed utility.
>
>    (iii.) The theoretical results are different. Our approach has both safety bound and utility bound while DeepStack argued that Method 1 enjoyed safety bound while Method 2 did not have any bound.

---

> > ### Comment · Reviewer_ByZa · 2022-08-07
> > **Response to response**
> >
> > Thank you for the response. What I had in mind is Method 2. It does appear there is a difference between Method 2 in DeepStack and the method described in this paper. Am I correct that the difference is that this paper uses the "Maxmargin" search whereas Method 2 in DeepStack uses the standard "resolving" search?

---

> > > ### Author Response · Authors · 2022-08-08
> > > **Response to ByZa**
> > >
> > > Right, one difference in technical details is that our method uses "Maxmargin" search, while DeepStack uses "resolving". Another fundamental difference is that our method studies the problem of safety and exploitation of the opponent's weakness while DeepStack focuses on accelerating the subgame resolving process. In addition, the theoretical results (safety and utility bound VS no theoretical guarantee) and the empirical results (utility against different agents with different estimation error, comparison to RNR, mixing strategy...) of our method are also totally different from those of DeepStack.

---

### Official Review · Reviewer_Zzn9 · 2022-07-13

**Rating:** 7
**Confidence:** 3
**Soundness:** 4 excellent
**Presentation:** 3 good
**Contribution:** 3 good

**Summary:**

The authors introduce a novel algorithm, Safe Exploitation Search (SES), that allows trading off safety and opponent exploitability in imperfect information zero-sum games. SES comes with theoretical upper bounds for exploitability, and lower bounds for performance. Unlike previous approaches, SES can be efficiently computed in an online fashion. The authors complement their theoretical investigation by a series of experiments using opponents of varying strength on two different poker variants.


**Questions:**

* I am mildly confused by your use of $v$, which is defined conditioned on both an infoset AND action, but the action is later frequently omitted (such as in Equation (2)). What happens to the action?

* How does the uncertainty arising from opponent modeling propagate into SES? Can SES be modified to take uncertainty over the opponent model into account?



**Limitations:**

The authors do a good job of discussing both limitations of their work, as well potential negative social impact.

# raised score to 7 after rebuttal

**Strengths And Weaknesses:**

# Strengths

The paper is well-written and organised, the authors' claims and results are clearly presented and, to the best of my assessment, sound. SES is addressing a well-motivated need, and is a useful contribution to the community.

# Weaknesses

* The paper should be spell-checked - there are quite a few minor issues.

---

> ### Author Response · Authors · 2022-07-30
> **Response to reviewer Zzn9**
>
> We appreciate the reviewer for recognizing the motivation of the  addressed problem, as well as the writing quality of the paper.
>
> 1. **Q**: Definition of $v_p^\sigma(I)$.
>
>    **A**: We overload the notation of $v$ a little bit, and we thank the reviewer for pointing this out. The definition is $v_p^\sigma(I):=\sum_a v_p^\sigma(I,a)\pi^\sigma_p(I,a)$. We already explicitly define it in the latest version.
>
> 2. **Q**: How does the uncertainty arising from opponent modeling propagate into SES? Can SES be modified to take uncertainty over the opponent model into account?
>
>    **A**: Currently, SES takes the modeling error of the opponent policy into account. In Theorem 4.2, we define $\epsilon=\|\|p-\hat p\|\|_1$ as the L1 distance between the true distribution of the opponent and our estimation. It will appear as a $-\epsilon\eta<0$ term in the final lower bound. It means that the larger the modeling error is, the smaller the lower bound of the expected payoff is. In Figure 2, we also conduct experiments on different $\epsilon$.
>
>    A straightforward way to take uncertainty over the opponent model into consideration is to adapt $\alpha$ based on confidence. An agent can choose a large $\alpha$ while he/she is confident that the opponent model is accurate, while picking a small $\alpha$ for conservativeness. In practice, it is difficult to calculate the uncertainty of an estimation of an unknown strategy. Therefore, we can use a no-regret learning algorithm to tune $\alpha$ in an online fashion. Please refer to Appendix H for more information.

---

> > ### Comment · Reviewer_Zzn9 · 2022-08-07
> > **Convincing rebuttal.**
> >
> > Dear Authors,
> >
> > Many thanks for your rebuttal. I have reconsidered your submission carefully, and am now raising my score to a 7 - meaning I will advocate this paper to be accepted.
> > Best wishes,
> >
> > Reviewer Zzn9

---

> > > ### Author Response · Authors · 2022-08-08
> > > **Response to Zzn9**
> > >
> > > Thank you very much for your prompt feedback and raising the score to 7. But the current score is still 6 and we will really appreciate it if you can update the score accordingly.

---

### Author Response · Authors · 2022-08-07
**General Response**

Dear Reviewers,

Thank you very much again for your helpful comments, and all your time for reviewing our paper. We were wondering if there is anything else you would like to discuss, and/or if we could further improve the paper. We would very much like to engage with you in our responses to your questions/comments. If you have any remaining questions after reading our response, please feel free to post them here, and we would be more than happy to discuss them further.

Best Regards,

the Authors.

---

### Meta-Review · Area_Chair_AuAu · 2022-08-25

**Recommendation:** Accept
**Confidence:** Less certain

**Metareview:**

This paper proposes an algorithm for searching for safe opponent exploitation strategies. The algorithm is based on alternating between a a safe max-margin search and an unsafe exploitation search. The paper provides strong theoretical guarantees for the algorithm, and some experiments that show the advantages of this algorithm. The reviewers found the paper well written and the idea of the algorithm interesting. A concern raised by one reviewer is the limited novelty of this work, since it is an application of DeepStack. Another reviewer shared this concern during the discussion, but argues that the paper still has merits because of the provided theoretical bounds and experiments.

**Award:**

No

---

### Decision · Program_Chairs · 2022-09-14

Accept